# Particle-associated denitrification is the primary source of N₂O in oxic coastal waters

Xianhui S. Wan [1,2] ✉, Hua-Xia Sheng[1], Li Liu [1], Hui Shen [1], Weiyi Tang[2], Wenbin Zou[1], Min N. Xu [3], Zhenzhen Zheng[3], Ehui Tan[3], Mingming Chen[1], Yao Zhang [1], Bess B. Ward [2] & Shuh-Ji Kao [1,3] ✉

The heavily human-perturbed coastal oceans are hotspots of nitrous oxide (N₂O) emission to the atmosphere. The processes underpinning the N₂O flux, however, remain poorly understood, leading to large uncertainties in assessing global N₂O budgets. Using a suite of nitrogen isotope labeling experiments, we show that multiple processes contribute to N₂O production throughout the estuarine-coastal gradient, sustaining intensive N₂O flux to the atmosphere. Unexpectedly, denitrification, rather than ammonia oxidation as previously assumed, constitutes the major source of N₂O in well-oxygenated coastal waters. Size-fractionated manipulation experiments with gene analysis further reveal niche partitioning of ammonia oxidizers and denitrifiers across the particle size spectrum; denitrification dominated on large particles and ammonia oxidizers on small particles. Total N₂O production rate increases with substrate and particle concentrations, suggesting a crucial interplay between nutrients and particles in controlling N₂O production. The controlling factors identified here may help understand climate feedback mechanisms between human activity and coastal oceans.

Estuaries and coastal zones function collectively as a bioreactor at the land ocean boundary where physical, biological and anthropogenic activities are intense. These ecosystems are vital to humans, sustaining nearly half of the world's population and providing invaluable ecosystem services[1]. Unfortunately, they are undergoing unprecedented environmental degradation due to multiple stresses from human activities (e.g., nutrient discharge, aquaculture, fishing etc.) superimposed on global change (e.g., ocean warming, acidification, deoxygenation etc.)[2,3]. The massive discharge of human-induced nutrients (nitrogen (N) in particular) and erosional sediments to the coastal ocean are among the most fundamental anthropogenic perturbations to the global coastal ecosystems. They exert profound impacts on local ecosystems (e.g., water quality, benthic environment) and global climate (e.g., greenhouse gas emissions)[4–7], leading to persistent consequences including eutrophication, harmful algal blooms, loss of biodiversity, and water

quality degradation, and are thus identified as major threats to human sustainability[8].

Numerous studies have explored the processes and rates of N cycling in the eutrophied coastal oceans, providing a valuable baseline understanding of the intense N dynamics in this critical system[6,9,10]. However, the pathways and production rates of nitrous oxide (N₂O), a potent greenhouse gas and ozone-depleting substance, which represents an important climate feedback of the marine N cycle to anthropogenic perturbation, are much less studied compared to other N cycling processes, and have been only recently investigated in a limited number of coastal sites such as the Chesapeake Bay[11], the Baltic Sea[12,13], and the inner shelf of the East China Sea and South China Sea[14,15]. This lack is probably due to the technical challenge of simultaneously disentangling the complex N₂O production processes and quantifying the rates in highly dynamic coastal systems. Consequently, the source and the air-sea flux of N₂O in the coastal oceans remains highly uncertain,

[1]College of Ocean and Earth Sciences, State Key Laboratory of Marine Environmental Science, Xiamen University, Xiamen 361102, China. [2]Department of Geosciences, Princeton University, Princeton, NJ 08544, USA. [3]State Key Laboratory of Marine Resource Utilization in South China Sea, Hainan University, Haikou 570208, China. ✉e-mail: xianhuiw@princeton.edu; sjkao@xmu.edu.cn

even though it is widely recognized that these eutrophied waters are usually hotspots of $N_2O$ emissions[16–18]. As the increasing trend of N fertilization is predicted to continue, $N_2O$ emissions are expected to further increase in the future[19,20], raising the urgency of better understanding of $N_2O$ sources for emission mitigation in these key regions[16–18].

Delivery of sediments from land to the coastal zone, on the other hand, represents another fundamental human perturbation to the coastal ocean[5,7,21]. The intensive human perturbation superimposed by climate change leads to dramatic changes in particle concentration and size spectrum in coastal oceans, particularly those influenced by large rivers[5]. However, the potential impacts of particle concentration on the N cycle, and how the particles of different sizes interact with substrate concentrations to influence $N_2O$ production, are unknown. Two major processes contributing to $N_2O$ production are aerobic ammonia oxidation and anaerobic denitrification (including bacterial nitrifier-denitrification) processes. Both processes are regulated by multiple environmental factors such as substrate, organic matter, redox status, etc., among which the dissolved oxygen (DO) concentration functions as the primary control: ammonia oxidation is considered the dominant process in the oxygenated environment and denitrification is restricted to the DO depleted water[22]. The ecological niche separation of these processes in oxygenated marine environments, however, is complicated by emerging evidence that microenvironments created by particles facilitate multiple N transformation processes and $N_2O$ production[23–25]. In the oxygenated and turbid coastal zones, low DO microenvironments associated with particles might thus cause an underestimation of the role of denitrification in $N_2O$ production, yet, the contribution of the anaerobic pathway to $N_2O$ has not been quantified.

Human-induced nutrient discharge and sediment delivery exert profound impacts on biogeochemistry and greenhouse gas flux in the global coastal ocean. Yet, the underlying mechanism remains largely unclear, underscoring the urgent need to expand our knowledge in understanding the causality of human activities and N biogeochemistry in this critical ecosystem. The Southeast and East Asia are home to one third of the global population. Land use practices associated with high population densities in the coastal areas lead to soil erosion and high anthropogenic N loading, resulting in the prominent features of high turbidity, hyper-eutrophication and intensive $N_2O$ emission in the estuaries and their adjacent coastal zones[26–28]. These regions provide a natural laboratory to study the effects of particle and substrate interactions on N cycling and the associated $N_2O$ generation processes, and can improve our understanding of the climate feedback to human perturbation in the global coastal ocean.

To address the key knowledge gaps on the sources and rates of $N_2O$ production in these emission hotspots, we investigated the distribution, flux and production rates of $N_2O$ spanning large environmental gradients in three large estuaries and the adjacent coastal zones along the coast of China (Supplementary Fig. 1). We show multiple $N_2O$ production processes and niche-partitioning of nitrification and denitrification along the particle spectrum, providing insights into the source of $N_2O$ and helping to constrain and predict global $N_2O$ emissions.

## Results and discussion
### Intensive $N_2O$ flux from the eutrophied and turbid coastal waters
A total of 107 samples from various depths at 60 stations were collected during six research cruises to the Changjiang Estuary and the adjacent East China Sea (hereafter referred to as CJE), the Jiulong Estuary and the adjacent Taiwan Strait (hereafter referred to as JLE), and the Pearl River Estuary (PRE) (See the Methods section, and Supplementary Fig. 1; Supplementary Table 1, 2 for detail). All the investigated estuaries and coastal zones contained high N concentrations (median total dissolved nitrogen (DIN) was 130, 153, and 8 µmol $L^{-1}$ in

the Pearl River Estuary (PRE), Jiulong Estuary (JLE), and Changjiang Estuary (CJE), respectively). DIN decreased seaward with increasing salinity, indicating dilution of the highly eutrophied estuarine water by seawater (Fig. 1a–c; Supplementary Fig. 2a; Supplementary Table 1). Ammonium ($NH_4^+$) concentration was highest in the freshwater end-member of the PRE and JLE and decreased rapidly with salinity. Prominent nitrite ($NO_2^-$) accumulations occurred in these eutrophied estuaries; the median in the PRE (13 µmol $L^{-1}$) and JLE (21 µmol $L^{-1}$) were at the highest end of measured $NO_2^-$ in the global marine system[29,30]. Nitrate ($NO_3^-$) was the dominant DIN species in all the investigated regions except for two samples in the PRE in 2013 and two samples in the CJE in 2015 (Supplementary Fig. 2b–d, Supplementary Fig. 3a–c). These results were consistent with previous observations of nutrient distribution in these systems across different years and seasons, demonstrating a persistent and ubiquitous eutrophic status along the coastal zone of China, despite the slight reduction in nutrient concentration in the research areas in recent years[2,31].

The estuaries also featured high particle content (Fig. 1d–f; Supplementary Fig. 3d–f). The median total suspended matter (TSM) was 36, 31, and 5 mg $L^{-1}$, and the corresponding particulate nitrogen (PN) was 11, 8, and 2 µmol $L^{-1}$ in the CJE, JLE and PRE, respectively, suggesting ubiquitous high turbidity and organic loading. Unlike the consistent DIN distribution pattern in relation to salinity, high concentrations of TSM and PN were also observed at the mixing zone of the estuaries, suggesting different controlling mechanisms for the dissolved and particulate phases of N in the coastal ocean. PN concentration was highest in the PRE, and TSM was highest in the JLE. Both PN and TSM were lowest in the CJE. However, the PN: TSM (weight ratio, g/g) values were higher in the CJE than PRE and JLE, partially due to the higher contribution of autochthonous organics in the CJE compared to the other estuaries[32].

DO, $N_2O$, and nutrients were all measured on the same sampling depths (Supplementary Table 1–3). The median of DO was 145, 147, and 183 µmol $L^{-1}$, corresponding to the median saturation of 63, 69, and 87% in the PRE, JLE, and CJE, respectively, and was independent of salinity and temperature. For example, the lowest DO concentrations were observed in both the low temperature upper PRE in 2013, and in the high temperature lower PRE in 2020, indicating complex physical-biological control of DO distribution (Fig. 1g–i; Supplementary Fig. 3g–i). Taking 2 mg $L^{-1}$ $O_2$ (equal to 62.5 µmol $L^{-1}$) as a threshold of hypoxia[33], only 6 of 107 measurements indicate hypoxia, thus, these estuaries were overall well-oxygenated during our investigation.

$N_2O$ had a spatial pattern similar to the DIN, decreasing from upstream to the ocean. The median concentration was 47, 56, 13 nmol $L^{-1}$ in the CJE, JLE and PRE, respectively (Fig. 1j–l; Supplementary Fig. 3j–l). The concentrations were consistently over-saturated compared to air equilibration. Accordingly, the median of $N_2O$ flux from the water to atmosphere was 27, 76, 11 µmol $m^{-2}$ $d^{-1}$ in the CJE, JLE and PRE, respectively. These values are at the highest end of the reported values in the estuarine and coastal zone globally[28,34,35], suggesting these eutrophied and turbid coastal waters are among the most intense $N_2O$ source regions. However, the large range (1-1592 µmol $m^{-2}$ $d^{-1}$) of $N_2O$ flux suggests substantial spatial-temporal variation and complex sources of $N_2O$ in these heavily human-perturbed systems.

Both $N_2O$ concentration and $N_2O$ flux were significantly correlated with DIN and DO, followed by PN, and the correlation was best fitted using exponential regression (Supplementary Fig. 4a–f). These results supported the broadly held idea that DIN substrates and DO concentrations, along with organic matter supply, are fundamental controls of $N_2O$ production in marine environments[28,35,36]. The relationships between $N_2O$ flux and DIN, DO and PN were less constrained compared to $N_2O$ concentration, suggesting a more complex regulation such as temperature, wind speed and air $N_2O$ concentration on $N_2O$ flux[37]. Furthermore, the exponential increase of $N_2O$ concentration and flux with the decrease of DO and increase of DIN and PN

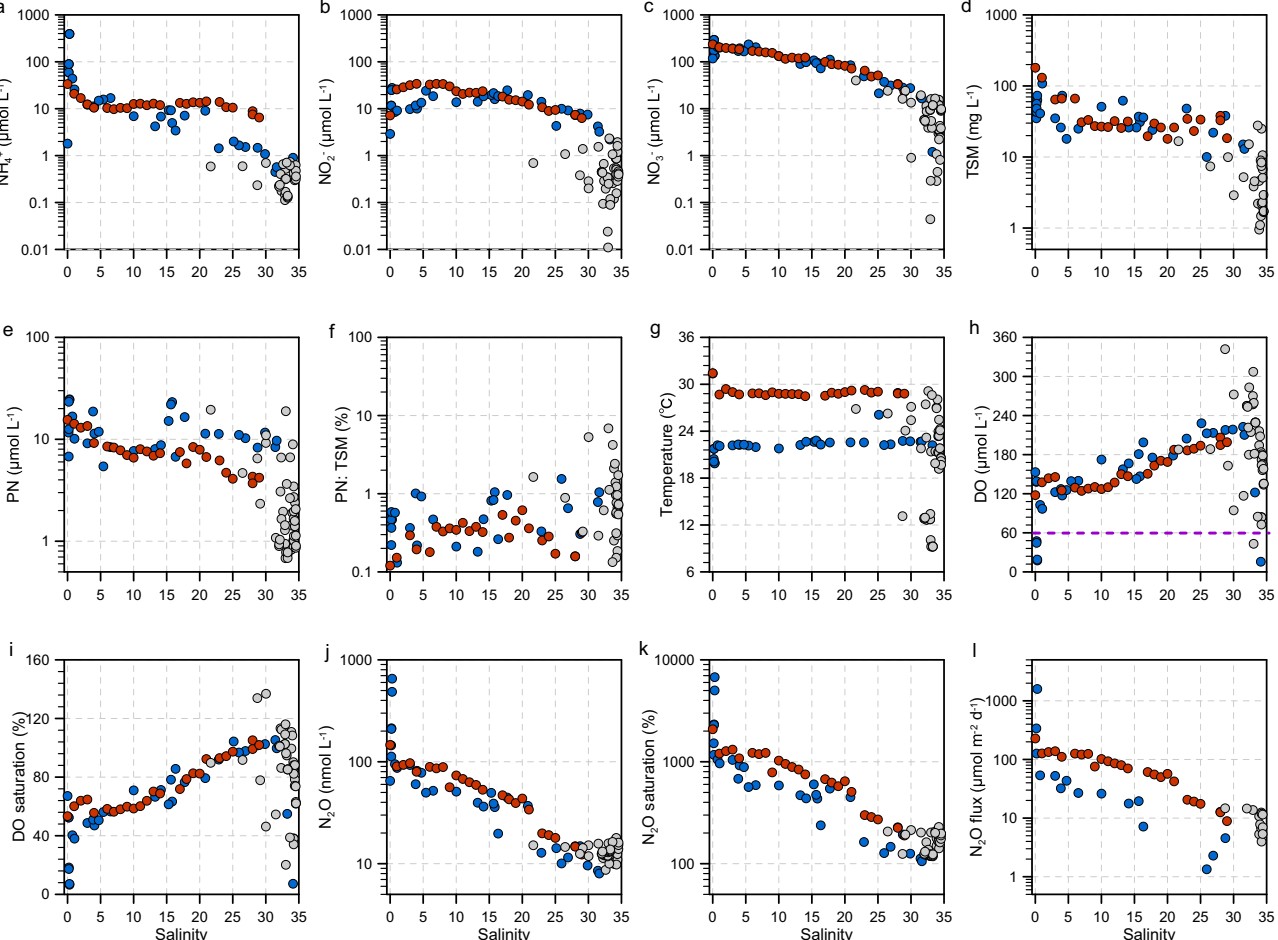

**Fig. 1 | Distribution of biogeochemical properties including nitrogen nutrients, dissolved oxygen (DO) and N₂O vs. salinity.** (**a**) ammonium (NH₄⁺); (**b**) nitrite (NO₂⁻); (**c**) nitrate (NO₃⁻); (**d**) total suspended matter (TSM); (**e**) particulate nitrogen (PN); (**f**) PN: TSM (weight ratio); (**g**) temperature; (**h**) DO; (**i**) DO saturation; (**j**) N₂O; (**k**) N₂O saturation; (**l**) N₂O flux. The blue, red and gray dots denote samples from the Pearl River Estuary (PRE) (2013 and 2020), the Jiulong Estuary (JLE) (2016 and 2018) and the Changjiang Estuary (CJE) (2015 and 2017), respectively. The purple dashed line in panel (**h**) denotes DO concentration of 62.5 µmol L⁻¹ (the threshold of hypoxia).

---

implied a nonlinear acceleration of N₂O emission during eutrophication and deoxygenation processes in the coastal ocean. These nonlinear relationships also suggest a potential bias if a simple fixed emission factor (e.g., N₂O: NO₃⁻ ratio) is used to derive N₂O budgets in the eutrophic coastal ocean. Given the wide prediction of increasing eutrophication and ocean deoxygenation in the global coastal ocean due to human activities and climate change[38], our results imply a positive feedback from the coastal ocean on global warming by enhancing N₂O emission.

## High N₂O production sustained by multiple N transformation pathways

A comprehensive set of isotope labeling incubation experiments was performed at selected stations, which exhibited large environmental gradients, to explore microbial N conversion and associated N₂O production rates. Like the DIN and N₂O distribution, the rates of ammonia oxidation, NO₂⁻ oxidation and NO₃⁻ reduction to NO₂⁻ showed a spatial distribution pattern that decreased from upstream to the ocean in the PRE and JLE. The rates were lower and the spatial pattern was less evident in the CJE compared to the PRE and JLE, likely because the CJE stations were distributed in the more open shelf region where the eutrophic fresh water was diluted by the nutrient-depleted ECS seawater. The sampling stations in the PRE and JLE were more concentrated upstream in the semi-closed bay area. Consequently, the CJE stations were characterized by higher and narrower

salinity range, and lower substrate concentration, leading to lower N conversion rates than the hyper-eutrophied PRE and JLE. Ammonia oxidation was detected in all measured samples and the rates varied over one to two orders of magnitude across the salinity gradient (Fig. 2a; Supplementary Fig. 5a; Supplementary Table 1). The rates measured in the PRE and JLE are at the higher end of the reported rates in the world's estuaries and coastal zones[39,40]. NO₂⁻ oxidation in the JLE and CJE was lower than in the PRE (Fig. 2b; Supplementary Fig. 5b). Consequently, the two steps of the nitrification process were more closely coupled in the PRE and CJE than in the JLE. Rates of NO₃⁻ reduction to NO₂⁻ were comparable between the JLE and PRE, and were lowest in the CJE (Fig. 2c; Supplementary Fig. 5c). NO₃⁻ reduction rate was lower than ammonia oxidation rate in all the investigated stations except for four samples in the PRE and JLE, indicating that ammonia oxidation is the main source of NO₂⁻ in these estuaries. Overall, the results showed that the aerobic ammonia oxidation was more active than anaerobic NO₃⁻ reduction, consistent with the fundamental control of DO on N dynamics in the marine environment.

N₂O production from NH₄⁺, NO₂⁻ and NO₃⁻ was detected at all stations. Similar to the distribution of N conversion rates, all the measured N₂O production rates showed a general seaward decreasing trend in the PRE and JLE, and the rates in these estuaries were higher than in the CJE (Fig. 2d–f; Supplementary Fig. 5d–f). Total N₂O production rate (sum of N₂O production from NH₄⁺, NO₂⁻ and NO₃⁻) ranged from 0.3–91.7, 0.4–58.1, and 0.1–2.5 nmol N L⁻¹ d⁻¹ in the PRE, JLE and CJE,

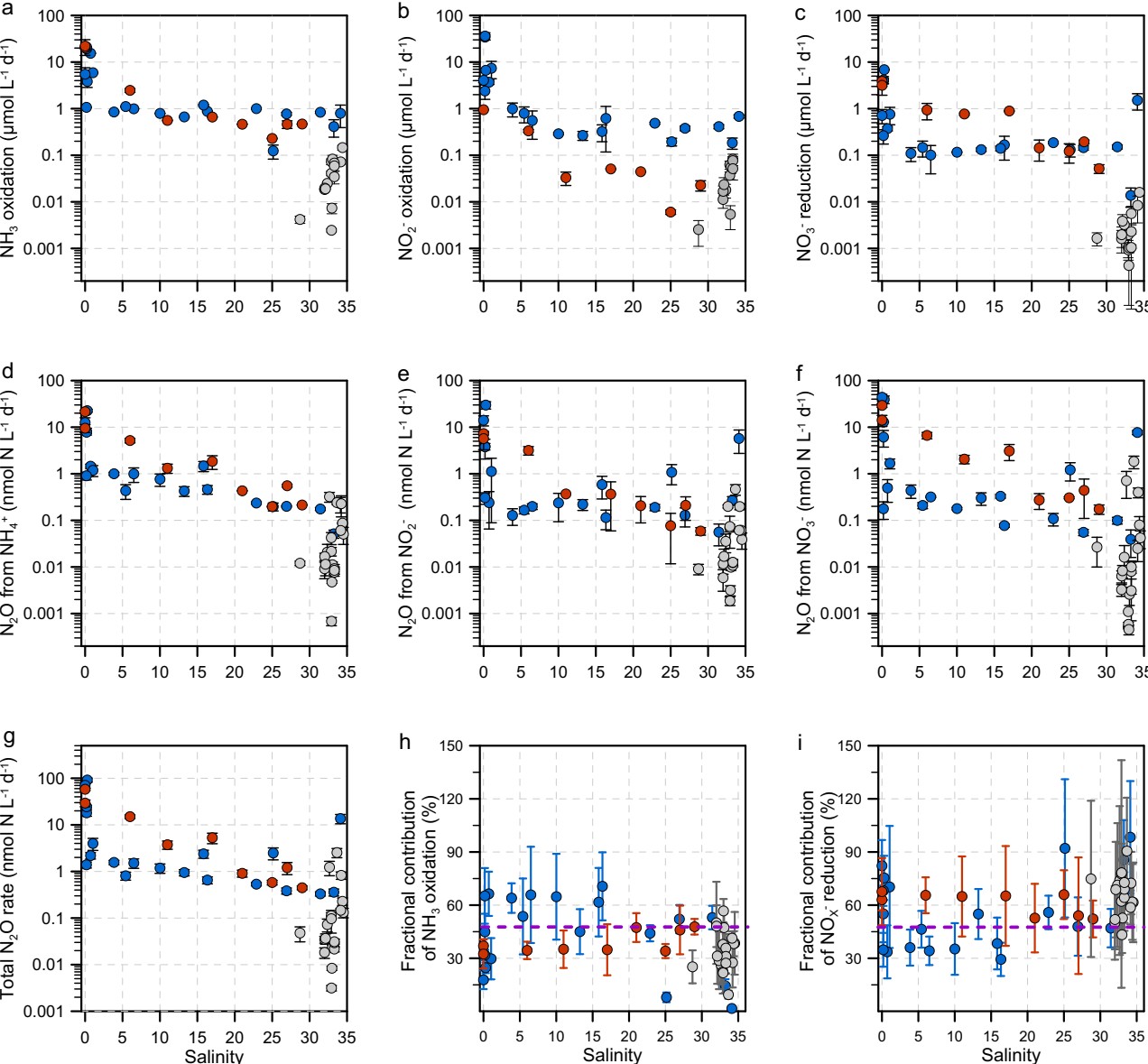

**Fig. 2 | Distribution of N conversion rates and N₂O production rates from multiple substrates.** (**a**) ammonia ($NH_3$) oxidation; (**b**) nitrite ($NO_2^-$) oxidation; (**c**) nitrate ($NO_3^-$) reduction to nitrite; (**d**) $N_2O$ production from ammonia oxidation; (**e**) $N_2O$ production from nitrite reduction; (**f**) $N_2O$ production from nitrate reduction; (**g**) total $N_2O$ production rate; (**h**) fractional contribution of $N_2O$ production from ammonia oxidation; (**i**) fractional contribution of $N_2O$ production from nitrite plus nitrate ($NO_X^-$) reduction. The blue, red and gray dots denote samples from the Pearl River Estuary (PRE), Jiulong Estuary (JLE) and Changjiang Estuary (CJE), respectively. Data are presented as mean rates ± standard deviation. Errors bars in panels (**a**–**g**) are standard deviation from triplicate incubations; errors bars in panels (**h**, **i**) are propagated standard deviations on the rates derived from triplicate incubations ($n = 3$ biologically independent samples). The purple dashed lines in panel (**h**) and (**i**) denote the value of 50%.

respectively (Fig. 2g; Supplementary Fig. 5g). The high N₂O production rate from all DIN species provided direct evidence that multiple biological N₂O sources sustain the high N₂O concentration and its subsequent emission to the atmosphere. Unexpectedly, N₂O from ammonia oxidation was frequently (37 out of the total 48 incubations) lower than from the reductive pathway (i.e., N₂O from $NO_2^-$ reduction plus $NO_3^-$ reduction), which accounted for more than half (median: 51, 64, and 70% in the PRE, JLE, and CJE, respectively) of the total N₂O production (Fig. 2h, i; Supplementary Fig. 5h, i). N₂O production from $NO_3^-$ alone accounted for nearly half of N₂O production in the investigated estuaries (Supplementary Fig. 6; Supplementary Note 1). These high rates of N₂O production from reductive pathways indicate a higher fractional contribution of N₂O from anaerobic N processes (partial denitrification) despite the fact that most of the stations were well-oxygenated and

ammonia oxidation rates were higher than $NO_3^-$ reduction rates. This result was contrary to the widely held perception that ammonia oxidation is the dominant source of N₂O in DO replete coastal zones[22,41], and indicated a more complex source structure for N₂O in oxygenated waters, which has not been adequately recognized.

N₂O yield (defined as N₂O production per $NO_2^-$ produced by oxidation of ammonia or reduction of $NO_3^-$) was consistently higher from $NO_3^-$ reduction than from ammonia oxidation in all three estuaries (Fig. 3a–c). N₂O yield from ammonia oxidation (median: 0.05, 0.11, 0.06% in the PRE, JLE and CJE, respectively) was at the lower range of reported values in the eutrophic coastal zones[42,43] and the open waters of North Pacific, North Atlantic and the Eastern Tropical Pacific[44–46]. N₂O yield from $NO_3^-$ reduction (median: 0.22, 0.34 and 0.25% in the PRE, JLE and CJE, respectively) was within the range

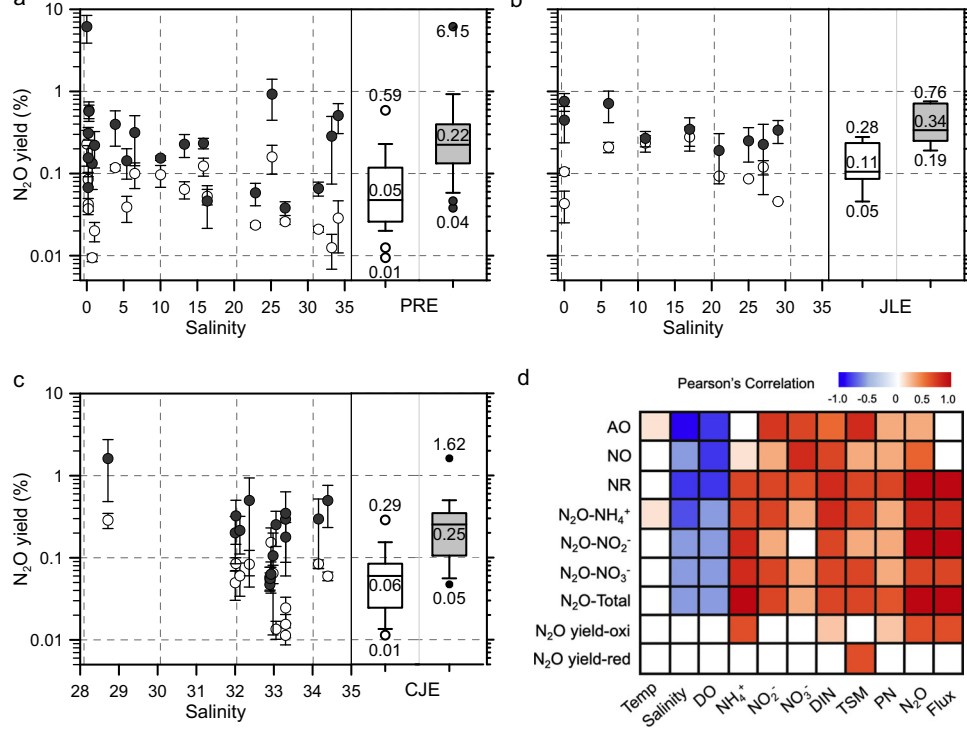

**Fig. 3 | N₂O yield during ammonia (NH₃) oxidation and nitrate (NO₃⁻) reduction, and Pearson's correlation between the measured rates and the environmental parameters.** (**a**–**c**) N₂O yield in the Pearl River Estuary (PRE), Jiulong Estuary (JLE) and Changjiang Estuary (CJE), respectively. The black and white dots denote N₂O yield during NO₃⁻ reduction and NH₃ oxidation, respectively. Data are presented as mean rates ± standard deviation. Errors bars in panels (**a**–**c**) are standard deviation on the rates derived from triplicate incubations (*n* = 3 biologically independent samples). The numbers in the box plots show the median value, and the minima and maxima values, respectively; whiskers and boxes show the 10% and 90% percentile and 25–75% quartile of the measurements, respectively (*n* = 20, 9, 19 stations in the PRE, JLE, and CJE, respectively). (**d**) The Pearson's correlation analysis for samples from all investigated areas (*n* = 48 biologically independent samples). Color gradient denotes the Pearson correlation coefficients. Only the significant correlations (*p* < 0.05, two sided) are colored, and the white squares indicate *p* > 0.05. AO, NO, NR, N₂O-NH₄⁺, N₂O-NO₂⁻, N₂O-NO₃⁻ denote the rates of ammonia oxidation, nitrite oxidation, nitrate reduction to nitrite, N₂O production from NH₄⁺, N₂O from NO₂⁻ and N₂O from NO₃⁻, respectively. N₂O total, N₂O yield-oxi, and N₂O yield-red mean the total N₂O production rate from NH₄⁺, NO₂⁻ and NO₃⁻, N₂O yield from ammonia oxidation, and N₂O yield during NO₃⁻ reduction, respectively.

reported in the main oxygen minimum zones and hypoxic coastal waters[11,46]. This higher N₂O yield during NO₃⁻ reduction is responsible for the higher N₂O production rate from partial denitrification, despite NO₃⁻ reduction rate being overall lower than ammonia oxidation rate. The N₂O yield of ammonia oxidation was positively correlated with the NH₄⁺ and DIN concentrations, indicating that a higher substrate concentration not only stimulates ammonia oxidation but also enhances N₂O production as a byproduct by increasing its yield (Fig. 3d). The yield of N₂O during NO₃⁻ reduction, on the other hand, was significantly correlated only with TSM, indicating a direct regulation of N₂O production efficiency by the particle concentration during partial denitrification in the turbid coastal waters. Ammonia oxidation is widely considered the main source of N₂O in oxygenated waters, and N₂O yield is widely used as a key component of biogeochemical model parametrization to assess and project marine N₂O production[47–49]. Our data demonstrate, however, that failure to include N₂O production from anaerobic N₂O production pathways, in addition to ammonia oxidation, and the varied yield under environmental gradients, would bias the estimation of N₂O sources and rates in the turbid coastal zone. These issues help explain the inadequacy of empirical or biogeochemical models to precisely estimate the N₂O budget in the coastal ocean and contribute to the large uncertainty in constraining N₂O flux in this globally relevant N₂O source region.

All six measured rates were negatively correlated with salinity, demonstrating a consistent spatial pattern of decreasing N conversion rates from the more eutrophied freshwater towards nutrient-poor seawater. The rates were all positively correlated with DIN, consistent with the fundamental regulation of N cycling rates by substrate availability (Fig. 3d). Although most of the rates were positively correlated with their specific substrate (e.g., NO₃⁻ reduction rate and NO₃⁻ concentration), ammonia oxidation was not strongly correlated with NH₄⁺ in the complete dataset. This relationship was driven by an anomalously low rate at station P01 at extremely high NH₄⁺ concentration under suboxic conditions in the PRE. No discernable interannual difference in the measured rates was found in any of the estuaries; likewise, no consistent relationship was found between the measured rates and temperature, likely because the cruises in different estuaries were carried out in different seasons, and the complex control of the microbial activities by multiple environmental factors such as substrates, DO and particle gradients, which may override the effect of temperature and mask the interannual variability. All the rates showed negative correlations with DO concentration. The increased rates for both N reduction and oxidation processes under lower DO concentration indicate intensified N recycling accompanied by a high DO consumption rate. Similar results have been widely observed in various marine systems, e.g., the highest N recycling rates are usually observed in the oxycline where the steepest DO gradient is located at both coastal and open oceans[50]. Only 6 out of 107 total samples (and 5 of 48 total incubations) had DO concentrations below the hypoxia threshold; however, the lowest DO concentration was 16 μmol L⁻¹, which is above the generally recognized threshold for denitrification (e.g., 2–10 μmol L⁻¹)[51]. These results implied that factors other than bulk oxygen concentration can facilitate active N reduction processes even in oxygenated coastal waters.

The measured rates were all positively correlated with TSM and PN (Fig. 3d), indicating a tight relationship between particle concentration and the strength of N cycling and $N_2O$ production. Previous studies have identified large redox gradients and low DO microenvironments associated with marine particles, creating multiple niches for various N transformation pathways, including nitrification and partial denitrification for $N_2O$ production[23,24]. Particulate matter is also rich in organic and inorganic substrates, providing valuable resources for both chemolithotrophic and heterotrophic metabolisms[49]. Accumulating molecular investigations show a positive correlation between particle concentration and the abundance of nitrifiers and denitrifiers in the coastal ocean[52,53]. These factors might substantially expand the niche of anaerobic metabolism to the oxygen-replete water and increase N removal from the global ocean[54]. Our results further disclosed a major role of anaerobic processes in contributing to $N_2O$ production in these oxygenated and turbid coastal waters. The measured $N_2O$ production rates also showed a strong relationship with $N_2O$ concentration and flux, suggesting the high $N_2O$ concentrations and fluxes were largely sustained by in-situ biological N recycling in the water column. Together, these results indicated a significant role of particles in regulating N recycling and $N_2O$ production pathways in heavily human-perturbed coastal systems. They also implicate microenvironments associated with particulate material in expanding the niche of denitrifiers, enabling them to be major contributors to $N_2O$ production and emission in oxygenated coastal waters.

We further compared the in-situ $N_2O$ production in the water column to the air-sea $N_2O$ flux measured in each station (Supplementary Table 4). The depth-integrated $N_2O$ production rate accounted for 4–10% (average 7%) of the air-sea $N_2O$ flux measured in the CJE; and the contribution increased to 17–114% (average 44%) in the PRE, indicating an increased role of water column $N_2O$ production in the more eutrophied coastal waters. Likewise, the ratio of depth-integrated $N_2O$ production rate to sedimentary $N_2O$ production was 21–58% (average 36%), and 8–201% (average 48%) in the CJE and PRE, respectively. Although large uncertainty remains in the estimation due to the relatively low vertical sampling resolution and the high heterogeneity of sedimentary $N_2O$ production rates[14,15], these results revealed a substantial role of water column $N_2O$ production in sustaining $N_2O$ concentration and its subsequent emission in the coastal ocean.

## Niche partitioning of N cycling by particle size

To better understand the critical role of particles in regulating N transformation and $N_2O$ production, we conducted size-fractionated and particle enrichment experiments at two stations in the JLE, which were characterized by large gradients of TSM size structure and concentration (Supplementary Fig. 7; see Methods for details). The recovery efficiency of the particulate matter was assessed by comparing in-situ PN and POC concentrations with the summed concentrations of all sizes. The impacts of manipulation on the investigated rates was examined by comparing the bulk rates (the rate measured using the in-situ water without manipulation) with the sum of size-fractionated rates. The results showed a high particulate recovery and limited impact on ammonia oxidation, while the manipulation caused significant decrease in $NO_3^-$ reduction rate (Supplementary Fig. 8; Supplementary Note 2).

N conversion and $N_2O$ production rates depended on particle size and rates were negligible in the filtered control samples. Ammonia oxidation and associated $N_2O$ production rates were highest in the size range of 3–20 μm at Station JL0, the upstream station (Fig. 4a; Supplementary Fig. 9a, d), and 0.2–3 μm at JL27, the downstream station (Fig. 4e; Supplementary Fig. 9g, j). The contribution of rates across the particle size distribution was consistent regardless of the concentration factor for particle enrichments. Consequently, $N_2O$ production from ammonia oxidation showed a unimodal pattern that peaked in a narrow range (i.e., over 80% of the $N_2O$ was produced by fine particles

(defined as the size range of 0.2–20 μm)), and decreased rapidly at large particle sizes (>20 μm). By comparison, rates of $NO_3^-$ reduction and $N_2O$ production from $NO_3^-$ and $NO_2^-$ were less strongly related to size; high rates of $N_2O$ production from $NO_2^-$ and $NO_3^-$ were measured in all three size fractions 0.2–160 μm at both upstream (Fig. 4b; Supplementary Fig. 9b, c, e) and downstream (Fig. 4f; Supplementary Fig. 9h, i, k) stations. Accordingly, $N_2O$ production from $NO_2^-$ and $NO_3^-$ reduction was more evenly distributed across the size spectrum compared to $N_2O$ from ammonia oxidation. All the investigated rates increased with particle concentration (Fig. 4; Supplementary Fig. 9), which is consistent with the significant correlation between POM concentrations and the bulk rates (Fig. 3d). Thus, N cycling intensity was stimulated by increased POM concentration, likely due to the increased organic supply and or higher abundances of microbes associated with the particles[55,56]. It should be noted, the higher measured rates on the small particles was due to the fact that the small particles (<20 μm) dominated the particle composition (i.e., small particles accounted for 82% and 76% of the total particle concentration in the upper and lower stations, respectively) (Supplementary Fig. 7a, b). We examined the distribution of $N_2O$ production rates normalized to PN content (normalized to per mol PN) at each size to compare the rate distribution along the size spectrum. Normalized rates are higher on large particles than small particles in sustaining $N_2O$ production from both $NH_3$ oxidation and $NO_X^-$ reduction, indicating the large particles are more favorable for both nitrifiers and denitrifiers to produce $N_2O$. The normalized $N_2O$ production rate from $NH_4^+$ was higher than from $NO_3^-$ in the small particles, and the opposite was true for large particles (Supplementary Fig. 10), indicating a niche partitioning between nitrifiers and denitrifiers along the size spectrum.

A further comparison of the fractional contribution from ammonia oxidation vs. $NO_X^-$ reduction to $N_2O$ production in each size class showed a clear transition of ammonia oxidation dominance at the fine size to $NO_X^-$ reduction dominance at the large particles. Specifically, over 60% of $N_2O$ was contributed by ammonia oxidation on the fine particles (Fig. 4c, g). In contrast, $NO_X^-$ reduction was the major process responsible for $N_2O$ production (>60%) for the large particles at both stations (Fig. 4d, h), suggesting that larger particles provide more favorable microenvironments for reductive $N_2O$ production. The relative distribution of $N_2O$ production across the particle size spectrum underlines the substantial contribution of $NO_X^-$ reduction to $N_2O$ production despite its fractional contribution likely being underestimated due to the loss of denitrifier activity during filtration and particle retrieval procedures.

Analogous to the bulk rates reported above (Fig. 2), the ammonia oxidation rate was higher than $NO_3^-$ reduction at both stations in the particle addition incubations (Supplementary Fig. 9). The difference between $N_2O$ production rates from the oxidative pathway and reductive pathway (Fig. 4) was less pronounced due to the lower $N_2O$ yield from ammonia oxidation than from the partial denitrification process (i.e., median of $N_2O$ yield during ammonia oxidation and $NO_3^-$ reduction were 0.03% and 0.2% at the upstream station; and were 0.05% and 0.09% at the downstream station, respectively) (Supplementary Fig. 9f, l). The $N_2O$ yield from the size fractionation experiment was lower than the bulk yield at the same stations (i.e., 0.45% and 0.23% at the upstream and downstream stations, respectively), probably due to the loss of $N_2O$ production via partial denitrification caused by the manipulation process (Supplementary Fig. 8). Thus, the rate measured in the manipulation experiment likely represents a conservative estimate of $N_2O$ production via $NO_2^-$ and $NO_3^-$ reduction.

In parallel, the abundance of bacterial and archaeal amoA, and bacterial nirS and nirK genes was quantified in the size-fractionated particles. Similar to the TSM distribution in each size, the investigated genes were overall more concentrated in the fine-size particles at both stations (Supplementary Fig. 11). Both bacterial and archaeal amoA genes were detected. Bacterial amoA genes accounted for a larger

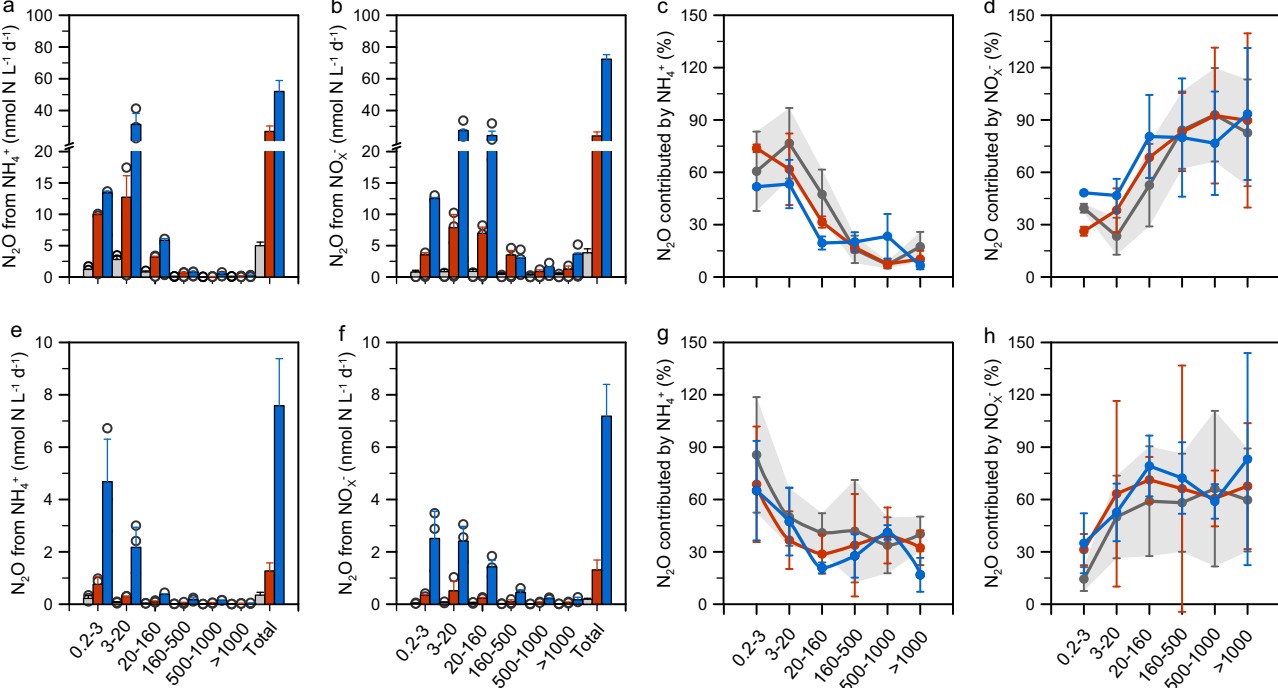

**Fig. 4 | Size-fractionated N conversion and $N_2O$ production rates in the Jiulong Estuary (JLE) (2018).** (**a–d**) rates measured at station JL0 (Salinity = 0), (**e–h**), rates measured at station JL27 (Salinity = 27). (**a, d**) $N_2O$ from ammonium ($NH_4^+$); (**b, f**) $N_2O$ from nitrite plus nitrate ($NO_x^-$); (**c, g**) fractional contribution of $N_2O$ production by ammonia oxidation at each size (i.e., rate of $N_2O$ from $NH_4^+$: total $N_2O$ rate at the corresponding size); (**d, h**) fractional contribution of $N_2O$ production by deni-trification (i.e., rate of $N_2O$ from $NO_x^-$: total $N_2O$ rate at the corresponding size). Total $N_2O$ rate is the sum of $N_2O$ production from $NH_4^+$, $NO_2^-$, and $NO_3^-$. At each station, six sizes of particles are retrieved by sequential filtration. Data are

presented as mean rates ± standard deviation. Errors bars in panels (**a, b, e, f**) are standard deviation from triplicate incubations ($n = 3$ biologically independent samples). The gray, red and blue bars and dots denote contribution derived from incubation with particle enrichment of 1, 5, and 10-fold compared to in-situ con-centration at the upstream station; and 1, 10, and 50-fold compared to in-situ concentration at the lower estuarine station. The gray shadows show the 95% confidence intervals derived from 1-fold particle concentration incubations. Error bars in panels (**c, d, g, h**) are propagated standard deviation on the rates derived from triplicate incubations ($n = 3$ biologically independent samples).

fraction in the upstream station, while the ammonia-oxidizing archaea (AOA) were more abundant at the downstream station. This result is consistent with the observation of ammonia-oxidizing bacteria (AOB) dominance in the more eutrophied conditions upstream and AOA dominance in less eutrophied seawater in many coastal zones globally[9,57]. The bacterial *nir* was dominated by *nirS* at both stations, and the total *nir* genes were more abundant than *amoA* genes in all the samples except for the 0.2-3 µm fraction at the downstream station (Supplementary Fig. 11a, d). Greater abundance of *nir* genes compared to *amoA* has also been widely observed in eutrophic estuaries, indi-cating a high N reduction potential in these highly perturbed coastal waters[11,58,59]. Moreover, our size-fraction manipulation revealed an elevated *nir*: *amoA* ratio in the large particles as compared to the fine-size, indicating an increasingly important role of denitrifiers in the large particles (Supplementary Fig. 11b, e). A consistent positive cor-relation between the fractional contribution of $N_2O$ from the $NO_x^-$ reduction and *nir*: *amoA* ratio was found, supporting the increased contribution to $N_2O$ production by N reduction relative to N oxidation in the larger particles (Supplementary Fig. 11c, f).

Another independent particle manipulation experiment was car-ried out at four stations in the PRE in 2020. For the bulk seawater, consistent with the result from the PRE in 2013, the ammonia and $NO_2^-$ oxidation rates were higher than $NO_3^-$ reduction rates at all stations except for the hypoxic A10 station (in-situ DO of 16 µmol $L^{-1}$) (Fig. 5a–c). $N_2O$ production rate from partial denitrification was higher than the rate via ammonia oxidation in all stations (Fig. 5d–f). For the <3 µm fraction (i.e., water from which particles over 3 µm had been removed by filtration), nearly all the investigated processes showed reduced rates, confirming the critical role of particles in sustaining N

conversion and $N_2O$ production. Rates of ammonia oxidation, $NO_2^-$ oxidation and $N_2O$ production via ammonia oxidation decreased by 35%, 41% and 21%, respectively, although the difference in rates between bulk and <3 µm samples was statistically significant only at station A1. By comparison, rates of $NO_3^-$ reduction, $N_2O$ production from $NO_2^-$ and $NO_3^-$ decreased by 79%, 96% and 92%, respectively, and the rates were significantly different at nearly all the stations, demonstrating greater reliance of denitrifiers than nitrifiers on parti-cles. These results were in good agreement with a similar size-fractionated manipulation performed in oxygen minimum zone waters of the Eastern Tropical North Pacific, showing that nitrification was less dependent on particles than were the N removal activities[56]. Thus, a change in particle size structure by human perturbation or extreme events might alter N conversion and $N_2O$ production rates even if the particle concentration remains unchanged. Together, these results revealed niche partitioning of ammonia oxidizers and denitrifiers on different sizes of particles. The source structures of $N_2O$ differ along the particle size spectrum, in which large particles sustain a substantial fraction of $N_2O$ production through the reductive pathways.

## Environmental implications

Coastal eutrophication was reported as early as a century ago, and the record has been expanding from western Europe to North America, then to East Asia. Eutrophication is predicted to occur in the rest of the world (e.g., South America and Africa) in the near future, becoming one of the most persistent and challenging global environmental issues[2,8]. While the global sediment loads have concomitantly increased[7] and a dramatic spatial change of sediment flux has been observed over the past decades[5], the influences of the rapid human-

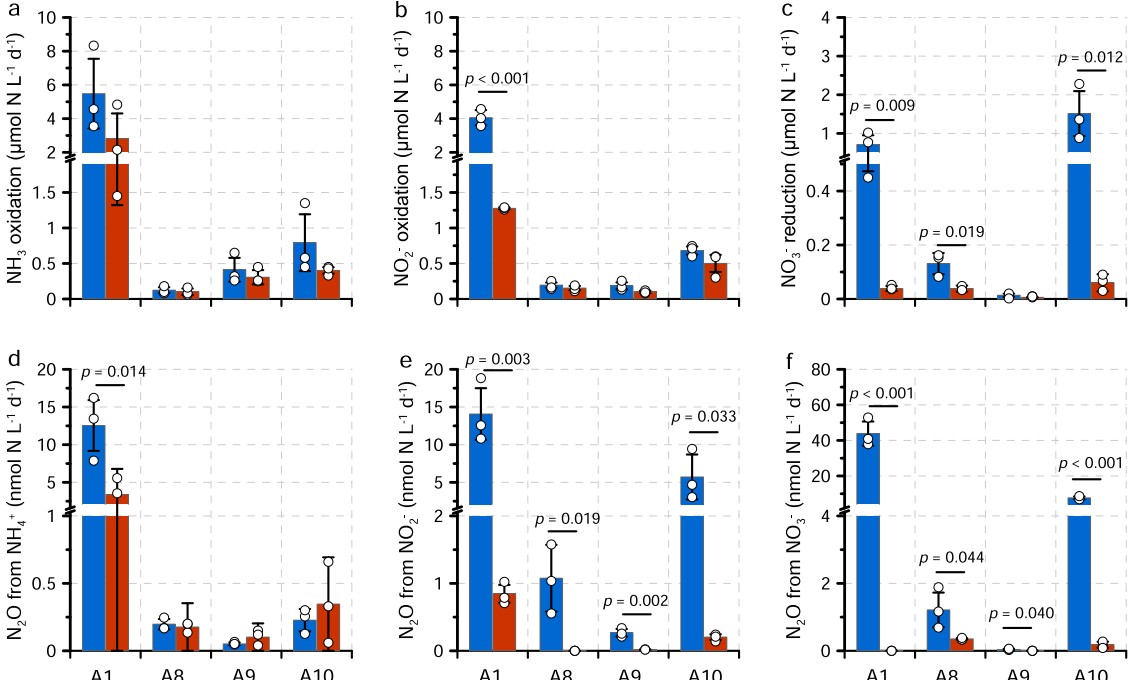

**Fig. 5 | Comparison of N conversion and N$_2$O production rates of in-situ water and particle-removed (>3 μm) samples at four stations (A1, A8, A9, A10) in the Pearl River Estuary (PRE) (2020).** The blue and red bars denote rates of bulk and particle-removed samples. (**a**–**f**) show the rates of ammonia oxidation, nitrite (NO$_2^-$) oxidation, nitrate (NO$_3^-$) reduction, N$_2$O from NH$_4^+$, N$_2$O from NO$_2^-$ and N$_2$O from NO$_3^-$, respectively. Data are presented as mean rates ± standard deviation. Errors bars are standard deviation from triplicate incubations ($n = 3$ biologically independent samples). The numbers show the significance levels when $p < 0.05$ ($t$ test, two sided).

and climate-induced changes of sediment delivery on marine biogeochemistry and N$_2$O production have not been recognized. Not only did our measurements support the general observation that estuaries can be large sources of N$_2$O emissions to the atmosphere, but they also revealed a crucial and interactive role for particles in controlling N cycling, in particular, the pathways and rates of N$_2$O production in highly eutrophied and turbid coastal waters (Fig. 6). Our study provides a mechanism for the conventional perception that eutrophied coastal zones are hot spots of N$_2$O emission to the atmosphere. The results further revealed in-situ N$_2$O production in the water column sustains 7% of air-sea N$_2$O flux in the CJE, and the ratio increased to 44% in the more eutrophied PRE, demonstrating a substantial role of the multiple N recycling pathways that contribute to the observed high N$_2$O emission. Moreover, the large contribution to N$_2$O production by partial denitrification associated with marine particles and the niche partitioning of nitrifiers and denitrifiers by particle size were previously underappreciated and poorly represented in biogeochemical models. Therefore, our findings should be helpful in improving the models that aim to constrain the global greenhouse gas budget and devise the best mitigation efforts.

The high contribution of denitrification to N$_2$O production observed in all investigated estuaries and seasons spanning large environmental gradients suggests a ubiquitous mechanism of the particle-associated microenvironment in sustaining anaerobic N$_2$O production in oxygenated waters. This finding has important implications for predicting N$_2$O emission in the emerging developing counties, where rapid development of eutrophication[20], and marked increase of suspended particle flux[5] due to urbanization and land use changes have been witnessed and are predicted to continue. Our results reveal a nonlinear increasing trend of N$_2$O production rate and varied N$_2$O source structure in response to exacerbated eutrophication and increased sediment delivery, identifying a positive feedback on warming by human activities; and offering understanding of the pathways and rates of N$_2$O production in the rapidly changing coastal ocean.

## Methods

### Field sampling and on-deck incubations

Samples were collected along the Pacific coastal zone of China during six research cruises conducted from 2013 to 2020 to the Changjiang Estuary and the adjacent East China Sea (hereafter referred to as CJE) (2015 and 2017), the Jiulong Estuary and the adjacent Taiwan Strait (hereafter referred to as JLE) (2016 and 2018), and the Pearl River Estuary (PRE) (2013 and 2020) (Supplementary Fig. 1). A total of 107 samples from various depths at 60 stations that spanned a wide range of temperature, salinity, DO, TSM, DIN (including NH$_4^+$, NO$_2^-$ and NO$_3^-$) and organic matter gradients were collected for N$_2$O distribution and flux measurements. On-board isotope labeling incubation experiments were performed at 48 depths for 31 selected stations (Supplementary Table 2).

Temperature and salinity were measured using the Seabird 911 CTD sensor package in the CJE and PRE cruises, and were measured using YSI6600D sensors in the JLE cruises. DO concentration was measured using the Winkler titration method in the PRE (2013) and CJE (2015) cruises, and using the electrode microsensor (Unisense, Denmark) in the remaining cruises. Discrete seawater samples were collected using twelve 12-liter Niskin bottles mounted to the CTD rosette in the CJE, and PRE cruises, and were collected using a 5-liter Perspex hydrophore water sampler in the JLE cruises.

Samples for chemical, biological and rate measurements were collected from the same casts. Triplicate 150-mL high-density polyethylene (HDPE) Nalgene bottles were used for nutrient collection; 250-mL glass serum bottles (Wheaton, USA) were used for subsequent N$_2$O measurements in the 2015 cruise, and 120-mL glass serum bottles (CNW, Germany) were used for N$_2$O sample collection in the remaining cruises. Seawater for subsequent analyses of TSM and PN was collected into 0.5 to 4-L polycarbonate Nalgene bottles depending on the particle content in the research area. All bottles and equipment were acid-washed and rinsed with in-situ seawater at least three times prior to sample collection. The glass bottles were pre-combusted at 450 °C for

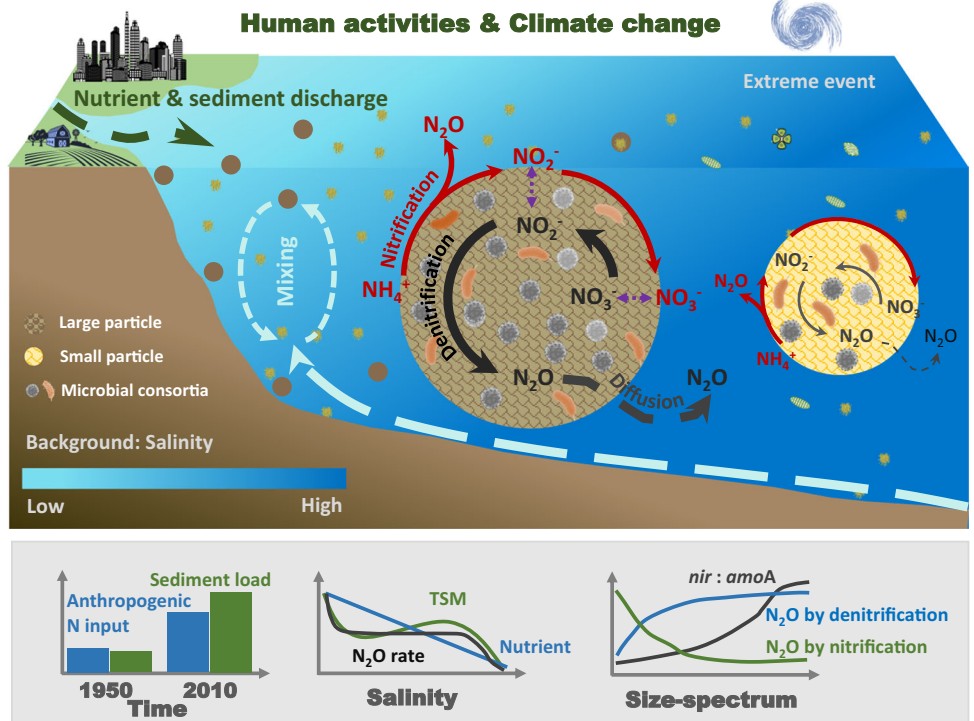

**Fig. 6 | Conceptual summary of particle-associated N$_2$O production pathways in the eutrophic and turbid coastal water.** The upper panel shows key physical (dashed lines) and biological (solid lines) processes in the hyper-eutrophied and high turbid estuarine and coastal waters. The brown circle and yellow circle in the figure denote nitrogen transformation and N$_2$O production pathways in the particle-associated microenvironment on the large and small particles, respectively. In both types of particles, the partial denitrification pathway contributes to a substantial source of N$_2$O in the well-oxygenated water, with the relative abundance of denitrifiers and contribution of particle-associated denitrification to N$_2$O increasing with particle size. The lower panel is the schematic illustration of global N nutrient input and sediment load over the past six decades (from 1950 to 2010)[5,20]; a cartoon of the spatial distribution pattern of nutrient concentration, total suspended matter (TSM) concentration and N$_2$O production rate against salinity in the investigated estuaries; and a cartoon of the distribution of *nir*: *amo*A, fractional contribution of denitrification and nitrification to N$_2$O production along the particle size-spectrum.

4 h before use. For N$_2$O concentration and incubations, serum bottles were overfilled two to three times before sealing with 20-mm butyl stoppers and aluminum crimp seals (Wheaton, USA). One mL of the sample was then removed using a syringe to allow HgCl$_2$ and tracer injection. Samples were preserved by adding 0.1–0.2 mL saturated HgCl$_2$ and were stored at 4 °C. For TSM and PN samples, seawater was gently (<200 mm Hg) filtered through a pre-combusted (450 °C for 4 h) and pre-weighed Whatman GF/F filter (25 mm diameter). After filtration, the filters were folded and wrapped in pre-combusted (450 °C for 4 h) aluminum foil and stored at −80 °C.

A comprehensive set of incubations was carried out using $^{15}$NH$_4$Cl (99% $^{15}$N, Cambridge), Na$^{15}$NO$_2$ (98% $^{15}$N, Cambridge) and K$^{15}$NO$_3^-$ (99% $^{15}$N, Cambridge) tracers in parallel to measure the N conversion (ammonia oxidation, NO$_2^-$ oxidation, NO$_3^-$ reduction) and N$_2$O production rates (Supplementary Table 3). All incubations were performed in the dark at near in situ temperatures (±2 °C) in 120-mL glass serum bottles, except on the 2020 PRE cruise when 60-mL glass serum bottles were used. Tracer from different stocks was injected (0.1–0.2 mL) into each bottle. The actual enrichment of the tracer was later calculated based on the measured NH$_4^+$, NO$_2^-$ and NO$_3^-$ concentration at each incubation depth. Overall, the final tracer concentration accounted for 38 ± 20%, 31 ± 24%, and 14 ± 18% of the final substrate pool in the $^{15}$NH$_4^+$, $^{15}$NO$_2^-$ and $^{15}$NO$_3^-$ labeling incubations, respectively. On the 2015 cruise, both N$_2$O production rate and N conversion rate were derived from the same incubation bottle. Immediately after tracer injection, -10 mL of the sample was replaced by pure N$_2$ and filtered through a 0.2 μm syringe filter to represent the initial condition (t0) for the N conversion rate incubations. The remaining water was preserved with 0.1 mL saturated HgCl$_2$ for N$_2$O

production measurement. The remaining bottles were incubated in the dark for 12 and 24 h on the 2015 cruise. At each timepoint, 10 mL of water was sampled from duplicate or triplicate bottles and then filtered for subsequent N conversion rate measurements, and the remaining water was preserved using HgCl$_2$ for subsequent determinations of N$_2$O. The filtrate was stored at −20 °C for subsequent analyses. During the remaining cruises, the N$_2$O production rate and the N conversion rate were incubated in independent serum bottles. The same procedures were used for onboard incubation, except that the N$_2$O rate incubation was terminated by adding 0.1 mL HgCl$_2$ without replacing by N$_2$, and the N conversion rate incubation was terminated by filtration through 0.2 μm PC filters. Time course incubations were conducted with five timepoints (0, 3, 6, 12, 24 h) in the 2013 PRE cruise; three timepoints (0, 12, 24 h) in the 2015 CJE cruise; three timepoints (0, 6, 12 h) in the 2016 JLE, 2017 CJE, and 2020 PRE cruises. All incubations were carried out in triplicate except the 2015 CJE cruise (duplicates at each timepoint).

Two sets of size-fractionated particle manipulation experiments were performed in the 2018 JLE and 2020 PRE cruises. In 2018, samples were collected at two stations in the upstream (Salinity = 0, JL0) and downstream (Salinity = 27, JL27) with distinct TSM size structure and concentration (Supplementary Fig. 7 a, b). Immediately after sample collection, the in-situ N$_2$O production and N conversion rate incubations were conducted following the method described above. The size-fractionated particles were collected for rate incubation and functional gene analysis by sequential filtration of seawater through nylon filters (-47 mm, 1000, 500, 160, 20 μm pore size) (Millipore, USA) and PC filters (47 mm, 3, 0.2 μm pore size) (Millipore, USA) via vacuum pump (<200 mm Hg) to produce a total of 6 size fractions for the incubations

(Supplementary Fig. 7c). For molecular samples, 5 L of water in the upstream station and 10 L in the downstream station were filtered by sequential filtration as described above. The particles on the nylon filters were then suspended using a Vortex (~1000 rpm for ~1 min) into 200 mL of particle free seawater and then filtered onto a 0.2 μm PC filter. For rate incubation samples, particle-free seawater at the corresponding site was prepared in parallel by filtration through PC disk filters (0.2 μm pore size) (Millipore, USA). The particle-free seawater was then dispensed into 250-mL HDPE bottles with 180 mL of water per bottle. One size-fractionated filter was added to each HDPE bottle, and the particles on the filter were released from the filters using a Vortex (~1000 rpm for 1–2 min). After removing the filter, the seawater with released particles was aliquoted into three 120-mL serum bottles (60 mL per bottle). Different particle concentration was achieved by adjusting the ratio of water used for particle collection and filtrate used for particle retrieval. In the upstream station (Salinity = 0), three particle concentrations (1, 5, and 10-fold of the in-situ concentration) were used for the incubation; the concentrations were set to 1, 10, and 50-fold of the in-situ concentration for the downstream station (Salinity = 27). Particle-free seawater (60-mL in 120-mL serum bottles) without particle additions was used as the control incubation. After the size-fractionated particles were dispensed into the particle-free seawater, $^{15}NH_4^+$, $^{15}NO_2^-$ and $^{15}NO_3^-$ tracers were introduced into the serum bottles to reach a final concentration of 5, 5, and 20 μmol L$^{-1}$, and 2, 2, and 10 μmol L$^{-1}$ for the upstream and downstream station, respectively (Supplementary Table 3). Immediately after tracer injection, ~5 mL of the sample was replaced by air and then filtered through a 0.2 μm syringe filter to represent the initial condition (t0) for the N conversion rate incubations, and the remaining water in the bottle was preserved with 0.1 mL saturated $HgCl_2$ for $N_2O$ production measurement. The remaining bottles were incubated in the dark for 3 and 6 h (triplicates at each timepoint). The filtrate for rate determination was stored at −20 °C for subsequent analyses. The PC filters for DNA analysis were stored at −80 °C until analysis.

In the 2020 PRE cruise, the size-fractionation experiment was carried out by removing particles (>3 μm) at four stations via inline filtration of seawater through 3 μm filters. The filtered seawater was dispensed into 60-mL serum bottles and purged with an $O_2$/He gas mixture to simulate the in-situ DO concentration. The incubation was conducted with three time points (0, 6, 12 h) with triplicates at each time.

## Nutrient, PN and N$_2$O concentration measurements
$NH_4^+$ concentration was measured on board the research vessels using the indophenol blue spectrophotometric procedure, with a detection limit of 0.5 μmol L$^{-1}$ (ref. [60]). $NO_3^-$ and $NO_2^-$ were measured using an AA3 Auto-Analyzer (Bran+Luebbe Co., Germany). The detection limits for $NO_x^-$ ($NO_3^-$ + $NO_2^-$) and $NO_2^-$ were 0.03 μmol L$^{-1}$ and 0.02 μmol L$^{-1}$, with precision better than 1% and 2.8%, respectively. TSM concentrations were obtained by dividing the dry weight of particles on the filters by the volume of water filtered. After TSM measurement, PN concentration was measured using a modified wet digestion method for samples collected from 2017 CJE cruise[61,62]. Briefly, the PN (with filters) was oxidized to $NO_3^-$ by using 1 mL of purified persulfate oxidizing reagent (POR) and 4 mL of deionized water (DIW) in a 12 mL 450 °C pre-combusted boro-silicate glass tube. $NO_3^-$ concentration after digestion was measured by chemiluminescence[63]. For the remaining four cruises, PN concentration was measured using an EA-IRMS (Thermo Finnigan Flash EA 2000 interfaced to a Delta V$^{PLUS}$ isotopic ratio mass spectrometer) system. The precision for PN concentration is <1%[62].

$N_2O$ concentrations were measured using two independent methods. During 2013, $N_2O$ concentrations were measured using a purge and trap system coupled with a gas chromatograph (Hewlett-Packard Model 6890 equipped with a micro-electron capture detector). Calibration of $N_2O$ concentrations was determined from peak areas with standard gases of 1.0 to 5.0 ppmv $N_2O$/$N_2$ (Research Institute of China National Standard Materials), which were run at 6 sample intervals. The precision of this method was estimated to be better than ±5% (ref. [25]). For the remaining cruises, $N_2O$ concentration was derived from ion peak area [m/z = 44] during isotope analysis using the GC-IRMS system (see below).

## Isotopic analyses of $NO_2^-$, $NO_3^-$ and $N_2O$
$δ^{15}N$ of $NO_2^-$ was measured by chemical conversion (sodium azide, Sigma-Aldrich) of $NO_2^-$ to $N_2O$[64]. For $δ^{15}N$ of $NO_3^-$ determination, the $NO_2^-$ was initially removed from samples by adding sulfamic acid (≥99% sulfamic acid, Sigma-Aldrich)[65] and the $δ^{15}N$ of $NO_3^-$ was determined using the bacterial denitrifier method[66]. Briefly, $NO_3^-$ was quantitatively converted to $N_2O$ using the bacterial strain *Pseudomonas aureofaciens* (ATCC No. 13985). The $δ^{15}N$ of $N_2O$ converted from $NO_2^-$ and $NO_3^-$ was measured using a Thermo Finnigan Gasbench system (including cryogenic extraction and purification) interfaced to a Delta V$^{PLUS}$ isotopic ratio mass spectrometer. $δ^{15}N$ of $NO_2^-$ values were calibrated against three in-house $NO_2^-$ standards ($δ^{15}N$ of the three in-house $NO_2^-$ standards were determined using the bacterial method, with values of 0.5 ± 0.3‰, 22.1 ± 0.5‰ and 96.3 ± 0.6‰, respectively). $δ^{15}N$ of $NO_3^-$ values were calibrated against $NO_3^-$ isotope standards USGS 34, IAEA N3 and USGS 32, which were run before, after, and at ten sample intervals. Accuracy (pooled standard deviation) was better than ±0.4‰ and ±0.2‰ for the chemical and bacterial methods, respectively.

Concentrations and isotopes of $N_2O$ were measured using a modified GC-IRMS (Thermo Finnigan Gasbench interfaced to a Delta V$^{PLUS}$ isotopic ratio mass spectrometer) with large volume purge and trap system[67]. Briefly, two needles were used for sample transfer and He pressurization; sample was transferred into a sparging flask (Pyrex, USA) using ultra-high purity He (>99.999%) and purged with He. Sample was purged for 30 min at a flow rate of 50 mL min$^{-1}$. The extracted gases were passed through an ethanol trap with dry ice and a chemical trap filled with magnesium perchlorate and Ascarite to remove $H_2O$ and $CO_2$. $N_2O$ was trapped by liquid nitrogen twice for purification and concentration and then injected into the GC-IRMS with He as carrier gas. $N_2O$ concentrations were determined by ion peak area [m/z = 44]. Calibration of $N_2O$ concentration was calculated from ion peak areas [m/z = 44] with standard gases of 199.6 and 501.0 ppmv $N_2O$/He (Research Institute of China National Standard Materials), which were run at ~10 sample intervals. The precision of this method was estimated to be better than ±3%. $δ^{15}N$ and $δ^{18}O$ were calibrated against two reference tanks (R1: 199.6 ppmv $N_2O$/He, $δ^{15}N$ = −3.2 ± 0.1‰ relative to air $N_2$, $δ^{18}O$ = 36.6 ± 0.1‰ relative to Vienna Standard Mean Ocean Water (VSMOW); R2: 501.0 ppmv $N_2O$/He, $δ^{15}N$ = −1.6 ± 0.1‰, $δ^{18}O$ = 36.6 ± 0.3‰). The precision of $δ^{15}N$ and $δ^{18}O$ measurements with 2 nmol $N_2O$ reference gas was better than 0.3‰ and 0.4‰, respectively[25].

## Functional gene abundance measurement
Four functional genes, bacterial *amo*A, archaeal *amo*A, bacterial *nir*S and *nir*K were quantified by quantitative PCR (qPCR) using a CFX96 Real-Time System (Bio–Rad Laboratories, Singapore) as described in Dai et al. (2022)[59]. The AOA shallow cluster was targeted with the primer set Arch-amoAFA and Arch-amoAR[68]. AOB are mostly affiliated with two groups: Betaproteobacteria (β-AOB) and Gammaproteobacteria (γ-AOB). Since the latter was below detection limit in estuaries along China[59], only β-AOB was targeted with the primer set amoA-1F and amoA-r New[69,70]. Bacterial *nir*S and *nir*K genes were quantified with the primer sets nirS-1F and nirS-3R[71] and nirK876 and nirK1040[72] (Supplementary Table 5). The presence of PCR inhibitors in DNA extracts was tested by qPCR with different dilutions of DNA (1-, 10-, and 100-fold dilutions). Based on these tests, we concluded that our

samples were not inhibited. Standard curves were constructed for the six genes using serial dilutions of quantified, linearized plasmid DNA from clone libraries generated from the PCR products. qPCR was performed in triplicate for each sample and negative controls without template were also included to test for contamination. The amplification efficiencies ranged from 90% to 100% with $R^2 > 0.99$ for each qPCR run. The specificity of qPCR products was verified by melting curves, agarose gel electrophoresis, and sequencing. The quantification limit of qPCR was one gene copy per reaction according to the estimates from maximum Ct-values generated by quantifiable samples (at least two of the three replicates were amplified).

### Surface N₂O saturation and air-sea flux

Surface $N_2O$ saturation was calculated using Eq. (1).

$$R = \frac{C_{obs}}{C_{eq}} \times 100 \tag{1}$$

where R (%) is the saturation of surface $N_2O$; $C_{obs}$ represents $N_2O$ concentration measured at the surface layer; $C_{eq}$ is the expected equilibrium concentration, which is computed using the Henry's Law[73] and the solubility dependence on temperature and salinity[74]. The air $N_2O$ concentration is the average atmospheric $N_2O$ concentration at Mauna Loa of the sampling month (NOAA/ESRL program).

Air-sea $N_2O$ flux was computed using Eqs. (2–3).

$$F = k \times (C_{obs} - C_{eq}) \tag{2}$$

$$k = 0.251 \times u^2 \times \left(\frac{S_c}{660}\right)^{-0.5} \tag{3}$$

where F ($\mu mol\,m^{-2}\,d^{-1}$) is air-sea flux of $N_2O$, k ($cm\,h^{-1}$) is the gas transfer velocity depending on wind and water temperature; u is mean wind speed at 10 m above sea surface during the cruise, as measured by the on-board meteorological station; Sc is the Schmidt number calculated from temperature[74].

### N conversion and N₂O production rates

$NO_2^-$ production from ammonia oxidation and $NO_3^-$ reduction was determined by the net accumulation of $^{15}N$ in the $NO_2^-$ pool during $^{15}NH_4^+$ and $^{15}NO_3^-$ labelling incubations. $NO_2^-$ oxidation rate was determined by the accumulation of $^{15}N$ in the $NO_3^-$ pool during $^{15}NO_2^-$ labelling incubations. For each process, the rate was computed based on Eqs. (4–6).

$$R_{AO} = \frac{d[^{15}NO_2^-]}{dt} \times \frac{[^{14}NH_4^+] + [^{15}NH_4^+]}{[^{15}NH_4^+]} \tag{4}$$

$$R_{NO} = \frac{d[^{15}NO_3^-]}{dt} \times \frac{[^{14}NO_2^-] + [^{15}NO_2^-]}{[^{15}NO_2^-]} \tag{5}$$

$$R_{NR} = \frac{d[^{15}NO_2^-]}{dt} \times \frac{[^{14}NO_3^-] + [^{15}NO_3^-]}{[^{15}NO_3^-]} \tag{6}$$

where $R_{AO}$, $R_{NO}$ and $R_{NR}$ are the ammonia oxidation, $NO_2^-$ oxidation and $NO_3^-$ reduction rate, respectively, t is the incubation time, $[^{15}NO_2^-]$ and $[^{15}NO_3^-]$ are the concentrations of $^{15}N$ in $NO_2^-$ and $NO_3^-$ pool in the sample, and $[^{14}NH_4^+]$, $[^{14}NO_2^-]$, $[^{14}NO_3^-]$ and $[^{15}NH_4^+]$, $[^{15}NO_2^-]$, $[^{15}NO_3^-]$ are the observed natural substrate concentrations and tracer concentrations, respectively.

Similarly, rates of $N_2O$ production from $NH_4^+$, $NO_2^-$ and $NO_3^-$ were derived based on the accumulation of $^{45}N_2O$ and $^{46}N_2O$. The production of $^{44}N_2O$ from a particular substrate was calculated according to

the $^{15}N$% after tracer addition (Supplementary Table 3). E.g., for $N_2O$ production by ammonia oxidation, $N_2O$ production rate was derived based on Eqs. (7–10).

$$R_{45N2O} = \frac{d[^{45}N_2O]}{dt} \tag{7}$$

$$R_{46N2O} = \frac{d[^{46}N_2O]}{dt} \tag{8}$$

$$R_{44N2O} = R_{46N2O} \times \frac{[^{14}NH_4^+] \times [^{14}NH_4^+]}{[^{15}NH_4^+] \times [^{15}NH_4^+]} \tag{9}$$

$$R_{N2O-NH4+} = R_{45N2O} + R_{46N2O} + R_{44N2O} \tag{10}$$

where $R_{44N2O}$, $R_{45N2O}$, $R_{46N2O}$ and $R_{N2O-NH4+}$ are $^{44}N_2O$, $^{45}N_2O$, $^{46}N_2O$ and gross $N_2O$ production rates during the $^{15}NH_4^+$ labeling incubation, respectively. $[^{45}N_2O]$ and $[^{46}N_2O]$ are the concentrations of measured $^{45}N_2O$ and $^{46}N_2O$. $N_2O$ production rates in $^{15}NO_2^-$ and $^{15}NO_3^-$ labeling incubation are calculated from the analogous equations. All rates were derived from the slopes of time course incubations with 2–3 replicates at each timepoint. For all six measured rates, the accumulation of $^{15}N$ product showed an overall significant linear increase over the time course incubation, suggesting a reliable rate estimation and minimal impact of any potential processes such as isotope dilution, $N_2O$ consumption, etc., in biasing the rate in the short-term incubations.

### Statistical analyses

The comparisons of reaction rates were examined by using the Student's t-test. A p-value of <0.05 was considered significant. The analyses were performed using SPSS Statistics 26.

### Reporting summary

Further information on research design is available in the Nature Portfolio Reporting Summary linked to this article.

## Data availability

All data needed to evaluate the conclusions in the paper are deposited in Zenodo database that can be accessed through https://doi.org/10.5281/zenodo.8092113.

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

## Acknowledgements

We greatly appreciate Y. Wu, M. He, X. Zhang, S. Wu's contributions during on board sampling and incubation in the research cruises. We also thank T. Huang, Q. Li for the on-board measurement of NH$_4^+$, L. Wang for NO$_3^-$ and NO$_2^-$ measurements and X. Yan for measuring the TSM, POC and PN concentrations. We are also grateful for the crew of the R/V *Ocean II*, R/V *Yuke* and R/V *Yanping II* for the onboard assistance and providing the CTD data. This work was supported by the National Natural Science Foundation of China through grants 92251306 and 41721005. X.S.W. and B.B.W. acknowledge funding from the Simons Foundation through award No. 675459 to B.B.W.

## Author contributions

X.S.W., B.B.W. and S.J.K. conceived the study and designed the experiment. X.S.W., H.X.S., L.L., H.S., W.Z., M.N.X., Z.Z., E.T., M.C. and Y.Z. performed the experiment and measured the samples. X.S.W., H.X.S., W.T., B.B.W. and S.J.K. analyzed the results and structured the manuscript. All authors contributed to the discussion of the results and editing of the manuscript.

## Competing interests

The authors declare no competing interests.
