## [Peer Review File · Nature Communications]

Particle-associated denitrification is the primary source of N₂O in oxic coastal watersREVIEWER COMMENTS

Reviewer #1 (Remarks to the Author):

In this study, authors have examined nitrous oxide production in estuary-coast gradient in China. The manuscript is nicely constructed and well-written, and the conclusions are supported by the work. I especially enjoyed reading the conclusive paragraph. The methodology is sound and well-described: authors have used stable isotope approach that allows them to separate and calculate the rates of three processes (ammonia oxidation, nitrite oxidation, nitrate reduction/denitrification) that can potentially produce N₂O, while many of the previous studies have only focused on measuring bulk N₂O concentrations or emissions in estuaries. In addition to measuring natural production rates, authors have also carried out a manipulation experiment to examine the role of particle size/quantity for N transformation processes at two sampling areas using stable isotope approach and quantification of functional genes, which is an interesting approach for explaining the dynamics between different processes.

Authors have collected an impressive dataset of more than a hundred samples representing various depths along the coastal gradient and several variables. However, data has been collected from six research cruises, each taking place in a different year and in PRE, there was 7 years between the cruises. Have authors checked how the measured variables varied between the years in the same sampling point? Would it be relevant?

The results show that N₂O concentration (and partly also N₂O fluxes) increase with increasing DIN and particulate N and decrease with increasing oxygen, pattern that is known based on the previous studies. The stable isotope incubation results show that in two out of three areas, the rates of the three processes decreased from upstream to ocean and that ammonia oxidation is the main source of nitrite and more active than denitrification. All three processes were found to produce N₂O, denitrification being the most important N₂O source, which is new knowledge and highlights the need to include denitrification in the N₂O budget estimates. All rates increased with increased particle concentration, but ammonia oxidation was found to be related to fine particles and NO_x reduction to larger particles, and this was further confirmed by gene abundance measurements. I think this is the most important and interesting finding of the study and provides new information on the dynamics between N transforming processes and should be highlighted even more in the abstract.

I have some minor comments/suggestions:

Line 33: What is “multiple stresses from human activities superimposed on global change”?

Line 38: How they are threat to sustainability?

Line 43: There are some recent studies on N₂O in estuaries/coastal ecosystems, e.g. Baltic sea (Aalto, S. L., Asmala, E., Jilbert, T., & Hietanen, S. (2021). Autochthonous organic matter promotes DNRA and suppresses N₂O production in sediments of the coastal Baltic Sea. *Estuarine, Coastal and Shelf Science*, 255, 107369., Hylén, A., Bonaglia, S., Robertson, E., Marzocchi, U., Kononets, M., & Hall, P. O. (2022). Enhanced benthic nitrous oxide and ammonium production after natural oxygenation of long-term anoxic sediments. *Limnology and Oceanography*, 67(2), 419-433.)

Line 53: What is meant with variable?

Line 60: But they depend on other factors also, like organic matter (denitrification)

Line 68: I would keep the manuscript on a more general level, no need to narrow it down to consider “only” Asia. I would assume that erosion/particle accumulation is a global problem?

Line 109: In which depth was DO measured? (Would be good to mention here as M&M comes after this section)

Line 147: It would be good to discuss on CJE more. Why rates were lower, if not related to these factors?

Line 227: Can the correlation between processes and TSM be also due to microbes bound to the particles? More or finer particles support higher microbial abundance (and activity?)?

Line 295: Why authors did not measure also nosZ abundance? It could be directly related to N₂O yield estimate

Line 449: I would remove here the information on gene analysis, as the sampling is explained detailed in lines 471-477.

Line 459: How were the different particle concentrations prepared? Was the concentration of particles measured after releasing them from filter?

Reviewer #2 (Remarks to the Author):

The manuscript looks at N cycling with regards N₂O in highly complex estuarine systems. The authors focus on suspended particles as the potential source of N₂O and investigate the contributions by inorganic nitrogen species. The manuscript is extremely comprehensive in this regard – three estuarine systems, a wide range of measurements coupled with fluxes is impressive. Furthermore, the manuscript is very well-written and presented. Some of the text for the subdivided figures is a little small, but its readable. The authors should be congratulated for putting this manuscript together.

The authors hypothesize that the N₂O results from nitrification on the outer part of a particle and denitrification on the inner side of a particle. The larger the particle, the higher opportunity for anaerobic conditions to exist in the center (as depicted in the Figure 6 schematic). However this trend is not observed in Figure 4 and some additional explanation is required. The questions related to this are:

- 1, Why are both nitrification and denitrification more prevalent in smaller particles? (Alternatively, why are they not prevalent in large particles)
- 2, Its not clear to me that opposing gradients of nitrification and denitrification are evident in Figure 4. Maybe I have missed something with the units of fractional contribution being used?
- 3, What happens if you have complete denitrification and N₂O consumption occurring. Under this scenario, you would not measure N₂O yet denitrification would be occurring.

The authors have done a good job with the tracer experiments and chemical measurements, but the methods used are best suited for whole water analyses. Since the emphasis is on micro-niches within particles then I would have prioritised with visualisation of the nitrifiers and denitrifiers using CARD-FISH. This would have provided evidence for the spatial distribution of these microbes as the manuscript predicts. Alternatively, the work could have measured chemical gradients within the particles using microsensors. For example O₂ concentrations can be measured at micrometer increments. The authors

cite two pieces of work (refs 18 and 19) which conducted these types of measurements.

The authors make a fairly generic claim that their datasets are important to constrain coastal N₂O emissions, but do not provide any evidence for this. A simple box model would indicate how much N₂O available for outgassing from the estuary derives from particles versus pelagic (non-particle) or the sediment. I realise the work did not include any sediment analyses, but maybe it exists in the literature or could be estimated? I think some quantitative prediction about the contribution from particles is required.

**Point by point response to reviewers' comments**

We thank the two anonymous reviewers for their constructive comments and
suggestions, which were very helpful in improving our manuscript. Below are our point
by point responses to the reviewers' comments (shown in blue), with the associated
changes in the manuscript (shown in red).

**REVIEWER COMMENTS**

**Reviewer #1 (Remarks to the Author):**

In this study, authors have examined nitrous oxide production in estuary-cost gradient
in China. The manuscript is nicely constructed and well-written, and the conclusions
are supported by the work. I especially enjoyed reading the conclusive paragraph. The
methodology is sound and well-described: authors have used stable isotope approach
that allows them to separate and calculate the rates of three processes (ammonia
oxidation, nitrite oxidation, nitrate reduction/denitrification) that can potentially
produce N₂O, while many of the previous studies have only focused on measuring bulk
N₂O concentrations or emissions in estuaries. In addition to measuring natural
production rates, authors have also carried out a manipulation experiment to examine
the role of particle size/quantity for N transformation processes at two sampling areas
using stable isotope approach and quantification of functional genes, which is an
interesting approach for explaining the dynamics between different processes.

**Response:** We are grateful for the reviewer's appreciation of our work, and we
appreciate that the reviewer recognizes the novelty and strength of our main findings
and the approaches used for the study.

Authors have collected an impressive dataset of more than a hundred samples
representing various depths along the coastal gradient and several variables. However,

data has been collected from six research cruises, each taking place in a different year
and in PRE, there was 7 years between the cruises. Have authors checked how the
measured variables varied between the years in the same sampling point? Would it be
relevant?

**Response:** The reviewer raised an important question regarding the interannual
variability and long-term trends of the water quality and nitrogen biogeochemistry in
the study area. Given the significance of estuarine-coastal systems to human
development, it is highly interesting to understand the evolution of this critical
ecosystem in response to human-perturbation and global change. However, identifying
the spatial-temporal variation of the coastal ocean remains a grand challenge due to the
highly dynamic nature and the complex relationship between human activity and
natural processes (e.g., Fennel and Testa, 2019; IOC-UNESCO, 2022). Several recent
review works have elaborately summarized the historical trajectory and current status
of nutrients and the associated biogeochemistry in the coastal ocean, highlighting the
complex and non-linear cause-effect relationship between the coastal ecosystem and
multiple stressors (e.g., Dai et al., 2023; Glibert and Burford, 2017; Winther et al.,
2020), which calls for long-term, multi-disciplinary research.

In our study, the six cruises were carried out in different seasons and years (PRE:
Autumn, 2013, and Summer, 2020; JLE: Summer, 2016, 2018; CJE: Spring, 2015, and
Summer, 2017). This temporal resolution is not sufficient to resolve interannual
variability and temporal trends of nitrogen biogeochemistry in the research area.

In response to the reviewer's comment, we compared the nitrogen nutrient and
conversion rate distribution between the PRE cruises (Fig. R1). The nitrogen nutrient
concentration in the 2020 cruise appeared to be lower than that in the 2013 cruise, with
two exceptions of NH_4^+ in the seawater end (Fig. R1a-c), indicating a reduction of
nitrogen input to the PRE due to enhanced sewage treatment and decreased fertilizer
use (Dai et al., 2023). However, it should also be noted that the variation could also be

caused by the significant seasonality in the PRE (Qian et al., 2022), or both. Unlike the
nitrogen nutrient concentration distribution, the nitrogen conversion rate (Fig. R1d-f)
and N₂O production rate (Fig. R1g-i) were not lower in 2020. The measured rates at the
stations with similar salinity were comparable between the two cruises, probably due
to the fact that the substrate was saturated for the nitrifiers and denitrifiers in both
cruises; thus, the reduction of substrate concentration in 2020 cruise probably had
limited impact on the nitrogen conversion and N₂O production rates. It is also likely
that the nitrogen conversion and N₂O production rates are co-regulated by multiple
environmental variables such as temperature, DO, organics, leading to more complex
response of the rates to substrate. This comparison reveals an overall limited impact of
nitrogen cycling and N₂O production despite reduction of nitrogen nutrient
concentration that occurred in recent years, calling for more extended and more
rigorous mitigation efforts to curb N₂O production.

Based on these analyses, we conclude that the comparison between the two
cruises is insufficient to resolve the interannual and long-term trends of nitrogen
biogeochemistry and N₂O production in the PRE. This is clearly a great question that
should be addressed in the future. In this revision, we made a few changes and added
one sentence to point out this important topic. Despite interannual variability and long-
term changes in particle or nutrient loading, the phenomenon of nitrous oxide
production from reductive processes occurred in the pelagic realm at all sites in all years.
The absolute rates would no doubt vary depending on interannual or long-term changes
in e.g., nitrogen delivery or particle loading, so we cannot extrapolate our rates too far
beyond the current study. But the process occurs widely across varying environmental
conditions and the finding that production from denitrification exceeds that from
oxidation is perhaps surprisingly consistent.

**Fig. R1 Nitrogen nutrient and conversion rates distribution in the PRE.** (a-c), NH_4^+ ,
 NO_2^- , and NO_3^- distribution. (d-f), Rates of ammonia oxidation, nitrite oxidation, and
 nitrate reduction to nitrite. (g-i), Rates of N_2O production from NH_4^+ , NO_2^- , and NO_3^- .
 The blue and red dots denote results from the 2013 and 2020 cruises, respectively.
 Errors bars in panels d-i are standard deviations from triplicate incubations.

References:

Dai, M. et al. Persistent eutrophication and hypoxia in the coastal ocean. *Cambridge*
 *Prisms: Coastal Futures* **1**, e19, 1-28 (2023).

Fennel, K. & Testa, J. M. Biogeochemical controls on coastal hypoxia. *Ann. Rev.*

*Mar. Sci.* **11**, 105-130 (2019).

Glibert, P.M. and Burford, M.A. Globally changing nutrient loads and harmful algal
blooms: Recent advances, new paradigms, and continuing challenges.
*Oceanography* **30**, 58–69 (2017).
IOC-UNESCO. State of the ocean report 2022: pilot edition. *IOC Technical Series*,
173 (edited by IOC, UN Educational, Scientific and Cultural Organization,
Paris) (2022).
Qian, W. et al. Long-term patterns of dissolved oxygen dynamics in the Pearl River
Estuary. *J. Geophys. Res. Biogeosci.* **127**, e2022JG006967 (2022).
Winther, J. G. et al. Integrated ocean management for a sustainable ocean economy.
*Nat. Ecol. & Evol.* **4**, 1451-1458 (2020).

**Revisions in the manuscript:**

Lines 113-116: These results were consistent with previous observations of nutrient
distribution in these systems across different years and seasons, demonstrating a
persistent and ubiquitous eutrophic status along the coastal zone of China, despite the
slight reduction in nutrient concentration in the research areas in recent years (e.g., 2,
31).

Lines 236-241: No discernable interannual difference in the measured rates was found
in any of the estuaries; likewise, no consistent relationship was found between the
measured rates and temperature, likely because the cruises in different estuaries were
carried out in different seasons, and the complex control of the microbial activities by
multiple environmental factors such as substrates, DO and particle gradients, which
may override the effect of temperature and mask the interannual variability.

The results show that N₂O concentration (and partly also N₂O fluxes) increase with
increasing DIN and particulate N and decrease with increasing oxygen, pattern that is
known based on the previous studies. The stable isotope incubation results show that in

two out of three areas, the rates of the three processes decreased from upstream to ocean
and that ammonia oxidation is the main source of nitrite and more active than
denitrification. All three processes were found to produce N₂O, denitrification being
the most important N₂O source, which is new knowledge and highlights the need to
include denitrification in the N₂O budget estimates. All rates increased with increased
particle concentration, but ammonia oxidation was found to be related to fine particles
and NO_x reduction to larger particles, and this was further confirmed by gene
abundance measurements. I think this is the most important and interesting finding of
the study and provides new information on the dynamics between N transforming
processes and should be highlighted even more in the abstract.

**Response:** The reviewer has nicely summarized the main findings of our study. We
thank the reviewer for the suggestion of improving the abstract by highlighting the new
findings about the contribution of partial denitrification to N₂O production and niche-
partitioning of nitrification and denitrification along the particle size spectrum. We
slightly changed the description and added a few words to highlight the main finding,
but we were not able to expand in detail due to the stringent word limit of the journal's
format requirement, i.e., no more than 150 words in the abstract.

**Revision in the manuscript:**

**Lines 21-23: Size-fractionated manipulation experiments with gene analysis further**
**reveal niche partitioning of ammonia oxidizers and denitrifiers across the particle size**
**spectrum; denitrification dominated on large particles and ammonia oxidizers on small**
**particles.**

I have some minor comments/suggestions:

Line 33: What is “multiple stresses from human activities superimposed on global
change”?

**Response:** The sentence has been revised for clarity.

**Revision in the manuscript:**

Lines 33-35: Unfortunately, they are undergoing unprecedented environmental
degradation due to multiple stresses from human activities (e.g., nutrient discharge,
aquaculture, fishing etc.) superimposed on global change (e.g., ocean warming,
acidification, deoxygenation etc.) (2-3).

Line 38: How they are threat to sustainability?

**Response:** We revised the sentence by adding more detailed information about the
detrimental effects of nutrient discharge and sediment erosion on the coastal ecosystem.

**Revision in the manuscript:**

Lines 38-42: They exert profound impacts on local ecosystems (e.g., water quality,
benthic environment) and global climate (e.g., greenhouse gas emissions) (4-7), leading
to persistent consequences including eutrophication, harmful algal blooms, loss of
biodiversity, and water quality degradation, and are thus identified as major threats to
human sustainability (8).

Line 43: There are some recent studies on N₂O in estuaries/coastal ecosystems, e.g.
Baltic sea (Aalto, S. L., Asmala, E., Jilbert, T., & Hietanen, S. (2021). Autochthonous
organic matter promotes DNRA and suppresses N₂O production in sediments of the
coastal Baltic Sea. *Estuarine, Coastal and Shelf Science*, 255, 107369., Hylén, A.,
Bonaglia, S., Robertson, E., Marzocchi, U., Kononets, M., & Hall, P. O. (2022).
Enhanced benthic nitrous oxide and ammonium production after natural oxygenation
of long-term anoxic sediments. *Limnology and Oceanography*, 67(2), 419-433.)

**Response:** We thank the reviewer for providing the literature information. The sentence
has been rewritten to include recent progress of N₂O pathways and rates study in the
coastal ocean.

**Revision in the manuscript:**

Lines 45-50: However, the pathways and production rates of nitrous oxide (N₂O), a
potent greenhouse gas and ozone-depleting substance, which represents an important
climate feedback of the marine N cycle to anthropogenic perturbation, are much less
studied compared to other N cycling processes, and have been only recently
investigated in a limited number of coastal sites such as the Chesapeake Bay (11), the
Baltic Sea (12, 13), and the inner shelf of the East China Sea and South China Sea (14,
15).

Line 53: What is meant with variable?

**Response:** The sentence has been re-written for clarity.

**Revision in the manuscript:**

Lines 59-62: The intensive human perturbation superimposed by climate change leads
to dramatic changes in particle concentration and size spectrum in coastal oceans,
particularly those influenced by large rivers (5).

Line 60: But they depend on other factors also, like organic matter (denitrification)

**Response:** The sentence has been revised to include other potential factors in regulating
the N₂O production processes.

**Revision in the manuscript:**

Lines 66-69: Both processes are regulated by multiple environmental factors such as
substrate, organic matter, redox status, etc., among which the dissolved oxygen (DO)
concentration functions as the primary control: ammonia oxidation is considered the
dominant process in the oxygenated environment and denitrification is restricted to the
DO depleted water (22).

Line 68: I would keep the manuscript on a more general level, no need to narrow it
down to consider “only” Asia. I would assume that erosion/particle accumulation is a
global problem?

**Response:** We agree with the reviewer that human-induced particle/ sediment
discharge to the coastal ocean is of global concern. Thus, our study holds a broader
implication for understanding the nitrogen biogeochemistry and N₂O production in the
global coastal ocean. We revised the description to emphasize our study area is an ideal
place for exploring this topic, but the results of the study have global significance.

**Revision in the manuscript:**

Lines 76-80: Human-induced nutrient discharge and sediment delivery exert profound
impacts on biogeochemistry and greenhouse gas flux in the global coastal ocean. Yet,
the underlying mechanism remains largely unclear, underscoring the urgent need to
expand our knowledge in understanding the causality of human activities and N
biogeochemistry in this critical ecosystem.

Lines 84-87: These regions provide a natural laboratory to study the effects of particle
and substrate interactions on N cycling and the associated N₂O generation processes,
and can improve our understanding of the climate feedback to human perturbation in
the global coastal ocean.

Line 109: In which depth was DO measured? (Would be good to mention here as M&M
comes after this section)

**Response:** DO concentration was measured in the same depths sampled for N₂O and
nutrients measurement, which can be found in Supplementary Table 1-3 and the raw
data set. The sample information has been added.

**Revision in the manuscript:**

**Lines 98-102:** A total of 107 samples from various depths at 60 stations were collected
during six research cruises to the Changjiang Estuary and the adjacent East China Sea
(hereafter referred to as CJE), the Jiulong Estuary and the adjacent Taiwan Strait
(hereafter referred to as JLE), and the Pearl River Estuary (PRE) (2013 and 2020) (See
the Method and materials section, and Supplementary Fig. 1; Supplementary Table 1,
2 for detail).

**Line 128-129:** DO, N₂O, and nutrients were all measured on the same sampling depths
(Supplementary Table 1-3).

**Line 147:** It would be good to discuss on CJE more. Why rates were lower, if not related
to these factors?

**Response:** The nitrogen conversion and N₂O production rates in the CJE were lower
than the rates measured in the other two estuaries. The reason was mainly due to the
difference in sampling station location, i.e., most of the stations sampled in the PRE
and JLE were distributed in the semi-closed estuarine zone with lower salinity (and thus
more freshwater with higher nutrient concentration), while for the CJE, the sampling
stations located at more open ECS shelf in the CJE cruises, and the freshwater from the
eutrophic Changjiang River was diluted by the relatively nutrient deplete ECS seawater,
which was evidenced by the overall higher salinity and lower nitrogen nutrient

concentration in the CJE (Fig. 1; Supplementary Fig. 1, 3). Given the fundamental
control of substrate concentrations on the measured rates, the relatively lower rates
observed in the CJE compared to JLE and PRE was primarily due to the overall lower
substrate concentrations in the CJE samples. In this revision, we expanded the
discussion to better clarify this point.

**Revision in the manuscript:**

**Lines 166-172:** The rates were lower and the spatial pattern was less evident in the CJE
compared to the PRE and JLE, likely because the CJE stations were distributed in the
more open shelf region where the eutrophic fresh water was diluted by the nutrient-
depleted ECS seawater. The sampling stations in the PRE and JLE were more
concentrated upstream in the semi-closed bay area. Consequently, the CJE stations
were characterized by higher and narrower salinity range, and lower substrate
concentration, leading to lower N conversion rates than the hyper-eutrophied PRE and
JLE.

Line 227: Can the correlation between processes and TSM be also due to microbes
bound to the particles? More or finer particles support higher microbial abundance (and
activity?)?

**Response:** We fully agree with the reviewer that the particle-associate microbes must
play an essential role in determining the observed relationship between the measured
rates and TSM concentration, which was further evidenced by our size-fractionated
experiment showing a similar trend of functional gene distribution and rate distribution
in each size. Marine particles are known as hotspots of microbial metabolism, providing
rich organic and inorganic substrates, and creating the microenvironment that enables
various nitrogen transformation pathways. We expanded the discussion of the
significant relationship between the measured rates and TSM.

**Revision in the manuscript:**

Lines 252-260: The measured rates were all positively correlated with TSM and PN
(Fig. 3d), indicating a tight relationship between particle concentration and the strength
of N cycling and N₂O production. Previous studies have identified large redox gradients
and low DO microenvironments associated with marine particles, creating multiple
niches for various nitrogen transformation pathways, including nitrification and partial
denitrification for N₂O production (23, 24). Particulate matter is also rich in organic
and inorganic substrates, providing valuable resources for both chemolithotrophic and
heterotrophic metabolisms (49). Accumulating molecular investigations show a
positive correlation between particle concentration and the abundance of nitrifiers and
denitrifiers in the coastal ocean (e.g., 52, 53).

Line 295: Why authors did not measure also *nosZ* abundance? It could be directly
related to N₂O yield estimate

**Response:** The reviewer is pointing out that the *nosZ* gene is another key functional
gene that directly links to N₂O cycling. Our study focuses on N₂O production pathways
and rates, however, so we focused on the functional genes related to N₂O production
rather than consumption.

Regarding the reviewer's concern about the bias of N₂O yield due to potential
N₂O consumption during the incubation, this potential impact (if any) should be
minimal because of the following reasons: 1) A recent investigation of the distribution
of N₂O-related functional genes shows the average *nir: nosZ* gene ratio is >10 our study
area, indicating only a small fraction of denitrifiers contains the *nosZ* gene (thus a low
N₂O consumption potential) in these eutrophic coastal waters (Dai et al., 2022). 2) In
the 2020 PRE cruise, we tested the N₂ production rate of the *in-situ* water, and we found
no detectable rate in all the ¹⁵NH₄⁺, ¹⁵NO₂⁻, and ¹⁵NO₃⁻ labeling incubations. The lack

of detectable N₂ production was likely caused by the relatively low sensitivity (because
of the very high *in-situ* N₂ concentration, i.e., > 1000 μmol N L⁻¹, which resulted in a
low detection limit of ~ 5 nmol N L⁻¹ d⁻¹ of N₂ production rate). Thus, we conclude the
N₂ production rate should be lower than 5 nmol N L⁻¹ d⁻¹, if any. Future experiments
by purging the *in-situ* N₂ could improve the sensitivity of N₂ production rate
measurement. We did not present this undetectable *in-situ* N₂ production rate in this
manuscript because any low N₂ production rate should not bias our N₂O production rate
estimation and N₂O consumption is not the main theme of the present study. 3) In light
of the relatively high *in-situ* N₂O concentration compared to the labeled N₂O production
in our short-term incubation, any potential N₂O reduction should largely consume the
*in-situ* rather the newly produced ¹⁵N₂O; and our result indeed showed strong linear
increase of ¹⁵N₂O production during the time course incubation (Fig. R2), suggesting a
reliable rate estimation during our time course incubation; and a minimal impact of any
potential N₂O consumption on our results. 4) Likewise, a ¹⁵NO₃⁻ labeled experiment
conducted in the CJE also finds no detectable N₂ production in the water column,
suggesting a low N₂ production by denitrification in the water column of the CJE (Yang
et al., 2022). For the reasons listed above, we conclude that *in-situ* N₂ production is low
and any effect of N₂O reduction on N₂O production rate estimates should be minimal
in our study.

Clearly, N₂O consumption is an important question but beyond the scope of the
present study. We added one sentence in the methods section to show the reliability of
N₂O production rate estimation.

**References:**

Dai, X. et al. Potential contributions of nitrifiers and denitrifiers to nitrous oxide
sources and sinks in China's estuarine and coastal areas. *Biogeosciences* **19**,
3757-3773 (2022).

Yang, J. et al. Sedimentary processes dominate nitrous oxide production and emission
 in the hypoxic zone off the Changjiang River estuary. *Sci. Total Environ.* **827**,
 154042 (2022).

 **Fig. R2** The linear increase of ¹⁵N₂O production during the ¹⁵NO₃⁻ labeled time-
 course incubation in selected stations in the 2103-PRE cruise. The significant linear
 increase of ¹⁵N-N₂O production suggests a reliable rate estimation in our incubation.

**Revision in the manuscript:**

Lines 646-650: For all six measured rates, the accumulation of ¹⁵N product showed an
 overall significant linear increase over the time course incubation, suggesting a reliable
 rate estimation and minimal impact of any potential processes such as isotope dilution,
 N₂O consumption, etc., in biasing the rate in the short-term incubations.

Line 449: I would remove here the information on gene analysis, as the sampling is
 explained detailed in lines 471-477.

**Response:** We revised the description of our size-fractionated filtration to reduce
redundancy.

**Revision in the manuscript:**

**Lines 504-507:** For molecular samples, 5 L of water in the upstream station and 10 L
in the downstream station were filtered by sequential filtration as described above. The
particles on the nylon filters were then suspended using a Vortex (~1000 rpm for ~1min)
into 200 mL of particle free seawater and then filtered onto a 0.2 μm PC filter.

Line 459: How were the different particle concentrations prepared? Was the
concentration of particles measured after releasing them from filter?

**Response:** The different concentrated particles were achieved by different ratios of
water used for particle collection to water used for incubation, e.g., 10 L of water was
used for filtration, the collected particles on the filters were retrieved to 1 L particle-
free water to get a 10-fold of the *in-situ* concentration. Due to the large water sample
required for the particle concentration experiment, i.e., 5 and 10-fold in the upper
stream station; 10 and 50-fold in the lower stream station, the efficiency of particle
retrieval and the impact of manipulation on the rates were only tested in 1-fold group
in each station. The results showed a high particulate recovery and limited impact on
ammonia oxidation, while the manipulation caused a certain decrease in NO_3^- reduction
rate (Supplementary Fig. 8; Supplementary Text 2). A sentence has been added to
clarify the procedure of particle enrichment.

**Revision in the manuscript:**

**Lines 513-515:** Different particle concentration was achieved by adjusting the ratio of
water used for particle collection and filtrate used for particle retrieval.

**Reviewer #2 (Remarks to the Author):**

The manuscript looks at N cycling with regards N₂O in highly complex estuarine
systems. The authors focus on suspended particles as the potential source of N₂O and
investigate the contributions by inorganic nitrogen species. The manuscript is
extremely comprehensive in this regard – three estuarine systems, a wide range of
measurements coupled with fluxes is impressive. Furthermore, the manuscript is very
well-written and presented. Some of the text for the subdivided figures is a little small,
but its readable. The authors should be congratulated for putting this manuscript
together.

**Response:** We are grateful for the reviewer's appreciation of the merit of our work,
and we appreciate that the reviewer recognizes the strength of our main findings and
the approaches used for the study. In this revision, we have carefully examined and
refined the figures to enhance their visibility.

The authors hypothesize that the N₂O results from nitrification on the outer part of a
particle and denitrification on the inner side of a particle. The larger the particle, the
higher opportunity for anaerobic conditions to exist in the center (as depicted in the
Figure 6 schematic). However this trend is not observed in Figure 4 and some additional
explanation is required. The questions related to this are:

1, Why are both nitrification and denitrification more prevalent in smaller particles?
(Alternatively, why are they not prevalent in large particles)

**Response:** As the reviewer noted, both nitrification and denitrification rates were
higher in the smaller particles (Fig.4 and Supplementary Fig. 9) than in the larger
particles. This result was largely due to the uneven distribution of particle concentration

at each size range, the small particles ($< 20 \mu\text{m}$) dominated the particle content (small
particles accounted for 82% and 76% of total particles in the upper and lower stations,
respectively) (Supplementary Fig. 7a, b). Following the reviewer's question, we
examined the distribution of N_2O production rates of normalized particle nitrogen
content (normalized to per mol PN) at each size. The results showed an overall higher
rate of large particles than small particles in sustaining N_2O production from both NH_4^+
oxidation and NO_x^- reduction, indicating the large particles are more favorable for both
nitrifiers and denitrifiers to produce N_2O . Moreover, the normalized N_2O production
rate from NH_4^+ was higher than from NO_3^- in the small particles, and the relationship
reversed in large particles (Supplementary Fig10). Consistently, we found a transition
of the dominance of N_2O production by NH_4^+ oxidation in small particles to NO_x^-
reduction dominance in large particles (Fig. 4c, d, g, h), suggesting a niche partitioning
of these nitrifiers and denitrifiers along the size spectrum.

Thus, the apparent dominance of both measured nitrification and denitrification
rates on small particles was due to the significantly higher concentration of small
particles than large particles in our sampling sites, which in turn masked the niche
partitioning (different prevalence) of nitrification and denitrification along the particle
size spectrum, which could be resolved by comparing the relative contribution of N_2O
production by nitrification and denitrification at each size; and by comparing the
normalized rates.

In this revision, we added a supplementary figure and expanded the associated
discussion to further clarify this important point.

**Revision in the manuscript:**

**Lines 310-321:**

It should be noted, the higher measured rates on the small particles was due to the fact
that the small particles ($< 20 \mu\text{m}$) dominated the particle composition (i.e., small
particles accounted for 82% and 76% of the total particle concentration in the upper

and lower stations, respectively) (Supplementary Fig. 7a, b). We examined the
 distribution of N_2O production rates normalized to particle nitrogen content
 (normalized to per mol PN) at each size to compare the rate distribution along the size
 spectrum. Normalized rates are higher on large particles than small particles in
 sustaining N_2O production from both NH_4^+ oxidation and NO_x^- reduction, indicating
 the large particles are more favorable for both nitrifiers and denitrifiers to produce N_2O .
 The normalized N_2O production rate from NH_4^+ was higher than from NO_3^- in the small
 particles, and the opposite was true for large particles (Supplementary Fig10),
 indicating a niche partitioning between nitrifiers and denitrifiers along the size
 spectrum.

Line 125-132 in the supplementary file:

**Supplementary Fig. 10 Normalized size-fractionated N₂O production rates in the**
**JLE (2018).** The rates are normalized to per mol PN at each size. a-b, rates measured
at upstream station JL0 (salinity=0); c-d, rates measured at upstream station JL27
(salinity=27). a, c: N₂O from NH₄⁺; b, d: N₂O from NO_x⁻. At each station, six sizes of
particles are retrieved by sequential filtration. Errors bars are standard deviation from
triplicate incubations. Note that the normalization was only performed in the 1-fold
particle retrieval manipulation, because size-fractionated PN concentration was
measured in this group.

2, Its not clear to me that opposing gradients of nitrification and denitrification are
evident in Figure 4. Maybe I have missed something with the units of fractional
contribution being used?

**Response:** As in response to the reviewer's last comment, the transition of nitrification
dominance to denitrification dominance along the particle size spectrum was likely
masked by the uneven distribution of particles. To overcome this issue, we normalized
N₂O production rates to per unit PN in each particle size, and found the niche
partitioning of these nitrifiers and denitrifiers along the size spectrum. Please see our
response to the last comment and the associated revision in the main text.

In this revision, we revised the description in the legend of Fig. 4 to better clarify
how the fractional contribution to N₂O was derived.

**Revision in the manuscript:**

**Lines 871-878: Fig. 4 Size-fractionated N conversion and N₂O production rates in**
**the Jiulong Estuary (JLE) (2018).** a-d, rates measured at station JL0 (Salinity=0), e-
483 h, rates measured at station JL27 (Salinity=27). a, d: N₂O from NH₄⁺; b, f: N₂O from
484 NO_x⁻; c, g: fractional contribution of N₂O production by ammonia oxidation at each
485 size (i.e., rate of N₂O from NH₄⁺: total N₂O rate at the corresponding size); d, h:

fractional contribution of N₂O production by denitrification (i.e., rate of N₂O from NO_x⁻:
total N₂O rate at the corresponding size). Total N₂O rate is the sum of N₂O production
from NH₄⁺, NO₂⁻, and NO₃⁻.

3, What happens if you have complete denitrification and N₂O consumption occurring.
Under this scenario, you would not measure N₂O yet denitrification would be occurring.

**Response:** Both reviewers raise the question regarding potential N₂O consumption
and how it would impact the observed production rates. As in response to Reviewer
#1, N₂O consumption (complete denitrification) is low in our study area and there is
evidence that the potential impact on N₂O production rate measurement is minimal:

1) Gene abundances indicate significantly higher N₂O production than N₂O
consumption potential in all of our investigated estuaries;

2) *the in-situ* N₂ production rate is below the detection limit in our study area,

3) the strong linear increase of ¹⁵N-N₂O in our time-course incubations
demonstrate a robust rate estimation. Please see our response to Reviewer #1 for a more
detailed discussion on this issue.

We added one sentence in the methods section to show the reliability of N₂O
production rate estimation.

**Revision in the manuscript:**

Lines 646-650: For all six measured rates, the accumulation of ¹⁵N product showed an
overall significant linear increase over the time course incubation, suggesting a reliable
rate estimation and minimal impact of any potential processes such as isotope dilution,
N₂O consumption, etc., in biasing the rate in the short-term incubations.

The authors have done a good job with the tracer experiments and chemical
measurements, but the methods used are best suited for whole water analyses. Since the

emphasis is on micro-niches within particles then I would have prioritised with
visualisation of the nitrifiers and denitrifiers using CARD-FISH. This would have
provided evidence for the spatial distribution of these microbes as the manuscript
predicts. Alternatively, the work could have measured chemical gradients within the
particles using microsensors. For example O₂ concentrations can be measured at
micrometer increments. The authors cite two pieces of work (refs 18 and 19) which
conducted these types of measurements.

**Response:** We are grateful for the reviewer's appreciation of our chemical
measurements and tracer labeling experiment, which represent the core part of our
study: quantifying the concentration, flux, and most importantly, the production
pathways and rates of N₂O in the coastal ocean.

We appreciate the appeal of using cutting-edge CARD-FISH and single-particle
microsensor technologies to probe the nuanced distribution of functional microbes and
redox gradients with ultra-high spatial resolution. Indeed, our work was partly inspired
by the accumulating molecular evidence of the cohabitation of nitrifiers and denitrifiers
in marine particles (e.g., Ganesh et al., 2015; Zhang et al., 2014; Zheng et al., 2022);
and the strong redox gradient measured within single marine aggregates (Klawonn et
al., 2015; Stief et al., 2016), as well as modeling work showing the particle associated
microenvironment expands substantially the niche of denitrifiers to the oxygenated
ocean (Bianchi et al., 2018). None of these studies, however, has addressed the
quantitative contribution of nitrification and denitrification to N₂O production
associated with the particles, leaving a large knowledge gap in understanding the
significance of N₂O production via denitrification in the oxygenated ocean. Thus the
main strength of our study lies in the quantification of denitrification in sustaining N₂O
production in the well-oxygenated but highly turbid coastal ocean. Our rate
measurements show that the mechanisms implied by the molecular localization and
modeling studies cited above really do result in N₂O production in oxygenated water.

Our main finding of the major contribution of partial denitrification in producing N₂O
and its subsequent flux to the atmosphere provides new insights on N₂O source
structure in these emission hotspots, which should also be conducive to model
development and global N₂O budget estimation.

We agree with the reviewer and other researchers who advocate the application
of state-of-the-art molecular biological technology such as CARD-FISH, ultra-high
resolution microsensor technology, isotope labeling incubation, and single-cell isotope
tracing (nano-SIMS), etc., to uncover more explicit information on niche separation of
nitrifiers and denitrifiers in particle-associated microenvironments, and to improve the
understanding on marine N₂O cycling (e.g., Wilson et al., 2020). The combined use of
these cutting-edge technologies might further enhance the strength of the main finding
of our study, but microscopic localization of microbes cannot be used to estimate actual
biogeochemical rates.

Since a high-resolution investigation of nitrifier and denitrifier distribution
within the particles is lacking in our study, we revised the schematic figure (Fig. 6) to
emphasize the pathways and rates of N₂O production in different sizes of particles,
which represent the primary finding of our study; and decrease the direct indications of
spatial distribution of the nitrifiers and denitrifies within a single particle.

**Reference:**

Bianchi, D., Weber, Thomas S., Kiko, R. & Deutsch, C. Global niche of marine
anaerobic metabolisms expanded by particle microenvironments. *Nat. Geosci.*
**11**, 263-268 (2018).

Ganesh, S., Parris, D. J., DeLong, E. F. & Stewart, F. J. Metagenomic analysis of
size-fractionated picoplankton in a marine oxygen minimum zone. *ISME J.* **8**,
187-211 (2014).

Klawonn, I., Bonaglia, S., Bruchert, V. & Ploug, H. Aerobic and anaerobic nitrogen
transformation processes in N₂-fixing cyanobacterial aggregates. *ISME J.* **9**,
1456-1466 (2015).

Stief, P., Kamp, A., Thamdrup, B. & Glud, R. N. Anaerobic nitrogen turnover by
sinking diatom aggregates at varying ambient oxygen levels. *Front. Microbiol.*
**7**, 98 (2016).

Wilson, S. T. et al. Ideas and perspectives: A strategic assessment of methane and
nitrous oxide measurements in the marine environment. *Biogeosciences* **17**,
5809-5828 (2020).

Zhang, Y., Xie, X., Jiao, N., Hsiao, S. S. Y. & Kao, S. J. Diversity and distribution of
*amoA*-type nitrifying and *nirS*-type denitrifying microbial communities in the
Yangtze River estuary. *Biogeosciences* **11**, 2131-2145 (2014).

Zheng, Y. et al. Overlooked contribution of water column to nitrogen removal in
estuarine turbidity maximum zone (TMZ). *Sci. Total Environ.* **788**, 147736
(2021).

**Revision in the manuscript:**

**Lines 895-909:**

**Fig. 6 Conceptual summary of particle-associated N₂O production pathways in the**
 **eutrophic and turbid coastal water.** The upper panel shows key physical (dashed
 lines) and biological (solid lines) processes in the hyper-eutrophied and high turbid
 estuarine and coastal waters. The brown circle and yellow circle in the figure denote
 nitrogen transformation and N₂O production pathways in the particle-associated
 microenvironment on the large and small particles, respectively. In both types of
 particles, the partial denitrification pathway contributes to a substantial source of N₂O
 in the well-oxygenated water, with the relative abundance of denitrifiers and
 contribution of particle-associated denitrification to N₂O increasing with particle size.
 The lower panel is the schematic illustration of global N nutrient input and sediment
 load over the past six decades (from 1950 to 2010) (5, 20); a cartoon of the spatial
 distribution pattern of nutrient concentration, TSM concentration and N₂O production
 rate against salinity in the investigated estuaries; and a cartoon of the distribution of

*nir: amoA*, fractional contribution of denitrification and nitrification to N₂O production
along the particle size-spectrum.

The authors make a fairly generic claim that their datasets are important to constrain
coastal N₂O emissions, but do not provide any evidence for this. A simple box model
would indicate how much N₂O available for outgassing from the estuary derives from
particles versus pelagic (non-particle) or the sediment. I realise the work did not include
any sediment analyses, but maybe it exists in the literature or could be estimated? I
think some quantitative prediction about the contribution from particles is required.

**Response:** We thank the reviewer for the constructive suggestion of quantifying the
contribution of *in-situ* N₂O production in the water column to air-sea N₂O flux, and the
importance of N₂O production in the water column compared to sediment, which is
very helpful in improving the significance of our study.

To address the reviewer's concern, we first computed the depth-integrated N₂O
production rate of the water column using trapezoidal extrapolation; note we excluded
those stations where only one depth was sampled because the vertical resolution was
too low to derive a reliable depth-integrated rate. Then, we calculated the contribution
of depth-integrated N₂O production rate to the N₂O flux measured at each station, and
we found that the depth-integrated N₂O production rate supported 4-10% (average 7%)
of the air-sea N₂O flux in the CJE; and the contribution increased to 17-114% (average
44%) in the PRE. Finally, we tried to extract the sedimentary N₂O production rate
investigated using intact cores in the same location from the literature. However, this
effort was hindered by the very limited number and huge heterogeneity of the measured
sedimentary N₂O production rates in the study area. As an alternative, we used the
average sedimentary N₂O production rate in the investigated estuaries, and found that
the ratio of depth-integrated N₂O production to sedimentary N₂O production rate
ranged from 21-58% (average 36%) in the CJE, and the ratio increased to 8-201%

(average 48%) in the PRE. Although this estimation remains uncertain due to our
limited vertical resolution in deriving depth-integrated rate, and the high spatial
heterogeneity of sedimentary N₂O production, these results suggested a substantial role
of N₂O production in the water column in sustaining N₂O emission in the coastal ocean,
particularly in the eutrophied estuarine area.

In this revision, we added a supplementary table and a paragraph discussing the
importance of water column N₂O production in the coastal ocean.

**Revision in the manuscript:**

Lines 271-281: We further compared the *in-situ* N₂O production in the water column
to the air-sea N₂O flux measured in each station. The depth-integrated N₂O production
rate accounted for 4-10% (average 7%) of the air-sea N₂O flux measured in the CJE;
and the contribution increased to 17-114% (average 44%) in the PRE, indicating an
increased role of water column N₂O production in the more eutrophied coastal waters.
Likewise, the ratio of depth-integrated N₂O production rate to sedimentary N₂O
production was 21-58% (average 36%), and 8-201% (average 48%) in the CJE and
PRE, respectively. Although large uncertainty remains in the estimation due to the
relatively low vertical sampling resolution and the high heterogeneity of sedimentary
N₂O production rates (14, 15), these results revealed a substantial role of water column
N₂O production in sustaining N₂O concentration and its subsequent emission in the
coastal ocean.

Lines 398-401: The results further revealed *in-situ* N₂O production in the water column
sustains 7% of air-sea N₂O flux in the CJE, and the ratio increased to 44% in the more
eutrophied PRE, demonstrating a substantial role of the multiple N recycling pathways
that contribute to the observed high N₂O emission.

Lines 152-160 in the supplementary file:

**Supplementary Table 4. Comparison of *in-situ* N₂O production in the water**
**column to air-sea N₂O flux and sedimentary N₂O production.** The depth-integrated
rate (from surface to bottom) was derived using trapezoidal extrapolation in stations
where multiple depths were sampled. Note we excluded those stations where only one
depth was sampled because the vertical resolution was too low to derive a reliable
depth-integrated rate. The sedimentary N₂O production rate was derived as the average
N₂O production rate from the intact core incubation from the literature.

Area	Station	WC N ₂ O rate ^a ($\mu\text{mol m}^{-2} \text{d}^{-1}$)	Flux ($\mu\text{mol m}^{-2} \text{d}^{-1}$)	WC N ₂ O rate: flux (%)	SD N ₂ O rate ^b ($\mu\text{mol m}^{-2} \text{d}^{-1}$)	WC N ₂ O rate: SD N ₂ O rate (%)
CJE	P1	1.5±0.8	14.7	10.3±5.6	2.6±2.5 (ref. 1)	57.9±31.4
CJE	C5	0.8±0.1	13.7	5.5±0.8	2.6±2.5 (ref. 1)	28.8±3.9
CJE	C3	0.5±0.2	14.6	3.7±1.3	2.6±2.5 (ref. 1)	20.5±7.1
PRE	A08	2.6±0.5	2.3	113.8±22.0	30.9±8.9 (ref. 2)	8.4±1.6
PRE	A06	2.4±1.2	7.2	33.6±5.2	30.9±8.9 (ref. 2)	7.8±1.2
PRE	A03	4.3±1.2	26.1	16.5±4.5	30.9±8.9 (ref. 2)	13.9±3.8
PRE	A01	16.6±6.2	32.2	51.5±19.3	30.9±8.9 (ref. 2)	53.6±20.0
PRE	P07	5.4±1.8	26.8	20.1±6.8	30.9±8.9 (ref. 2)	17.4±5.9
PRE	P05	10.3±5.4	53.5	19.3±10.0	30.9±8.9 (ref. 2)	33.3±17.3
PRE	P03	62.2±8.2	126.1	49.4±6.5	30.9±8.9 (ref. 2)	201.2±26.5

a: WC N₂O rate is the depth-integrated (surface to bottom) N₂O production rate of the water column.

b: SD N₂O rate is the sedimentary N₂O production rate measured using intact core.

REVIEWERS' COMMENTS

Reviewer #1 (Remarks to the Author):

I have read the previous version of the manuscript. I am happy to see that authors have answered all my concerns and questions. Already the previous version was of high quality, and after the revision, it is a pleasure to read this manuscript. It is well written, with very nice graphs and results that will provide new knowledge on particle-associated nitrous oxide production in coastal ecosystems.

Reviewer #2 (Remarks to the Author):

The authors have answered my questions satisfactorily

**Point by point response to reviewers' comments**

We thank the two anonymous reviewers for the appreciation of our work, and the
positive feedback regarding our revised manuscript. Below are our point by point
responses to the reviewers' comments (shown in blue).

**REVIEWER COMMENTS**

**Reviewer #1 (Remarks to the Author):**

I have read the previous version of the manuscript. I am happy to see that authors have
answered all my concerns and questions. Already the previous version was of high
quality, and after the revision, it is a pleasure to read this manuscript. It is well written,
with very nice graphs and results that will provide new knowledge on particle-
associated nitrous oxide production in coastal ecosystems.

**Response:** We are grateful for the reviewer's appreciation of our work, and we
appreciate the reviewer's support for the publication of our revised manuscript.

**Reviewer #2 (Remarks to the Author):**

The authors have answered my questions satisfactorily

**Response:** We thank the reviewer for the positive feedback on our revised manuscript.
